# *Amanita* Section *Phalloideae* Species in the Mediterranean Basin: Destroying Angels Reviewed

**DOI:** 10.3390/biology11050770

**Published:** 2022-05-18

**Authors:** Pablo Alvarado, Antonia Gasch-Illescas, Sylvie Morel, Magda Bou Dagher-Kharrat, Gabriel Moreno, José Luis Manjón, Xavier Carteret, Jean-Michel Bellanger, Sylvie Rapior, Matteo Gelardi, Pierre-Arthur Moreau

**Affiliations:** 1ALVALAB, Dr. Fernando Bongera St., Severo Ochoa Bldg. S1.04, E-33006 Oviedo, Spain; 2Departamento de Medicina, Universidad de Sevilla, San Fernando 4, E-41004 Sevilla, Spain; antonia.gasch.illescas@gmail.com; 3Institut Pasteur de Lille, 1 Rue du Professeur Calmette, 59000 Lille, France; 4Laboratory of Botany, Phytochemistry and Mycology, Faculty of Pharmacy, CEFE, CNRS, Univ Montpellier, EPHE, IRD, CS 14491, 15 Avenue Charles Flahault, CEDEX 5, 34093 Montpellier, France; sylvie.morel@umontpellier.fr (S.M.); sylvie.rapior@umontpellier.fr (S.R.); 5Laboratoire «Caractérisation Génomique des Plantes», Faculté des Sciences, Université Saint-Joseph de Beyrouth, Rue de Damas, BP 17-5208, Mar Mikhaël, Beyrouth 1104 2020, Lebanon; magda.boudagher@usj.edu.lb; 6Departamento de Ciencias de la Vida-Botánica, Universidad de Alcalá, E-28805 Alcalá de Henares, Spain; gabriel.moreno@uah.es (G.M.); josel.manjon@uah.es (J.L.M.); 7Independent Researcher, 68, Rue Alexis Maneyrol, 92370 Chaville, France; xavier.carteret@bbox.fr; 8CEFE, CNRS, Univ Montpellier, EPHE, IRD, INSERM, 1919 Route de Mende, CEDEX 5, 34293 Montpellier, France; jean-michel.bellanger@cefe.cnrs.fr; 9Associazione Micologica Ecologica Romana (AMER) APS, Via Tuscolana 548, I-00181 Rome, Italy; timal80@yahoo.it; 10Independent Researcher, Via dei Barattoli 3A, I-00061 Anguillara Sabazia, Italy; 11Faculté de Pharmacie, Université Lille, 3 rue du Pr Laguesse, 59000 Lille, France; pierre-arthur.moreau@univ-lille.fr; 12EA 4489 LGCgE, Université Lille, 59000 Lille, France

**Keywords:** *Agaricales*, amatoxins, phallotoxins, epidemiology, toxicology

## Abstract

**Simple Summary:**

Whitish lethal species of *Amanita* sect. *Phalloideae* (‘destroying angels’) are known to be among the most poisonous fungi worldwide due to their production of amatoxins. The taxonomy of species occurring in the Mediterranean region is here revised, clarifying the identity of several names. *Amanita decipiens*, *A. porrinensis,* and *A. virosa* var. *levipes* are here considered later heterotypic synonyms of *A. verna*, *A. phalloides,* and *A. amerivirosa*, respectively, while a new name, *A. vidua*, is proposed for a spring-occurring taxon. The amatoxins and phallotoxins present in Mediterranean destroying angels were characterized, and their epidemiology discussed on the basis of the case study of available data from Spain.

**Abstract:**

In Europe, amatoxin-containing mushrooms are responsible for most of the deadly poisonings caused by macrofungi. The present work presents a multidisciplinary revision of the European species of *Amanita* sect. *Phalloideae* based on morphology, phylogeny, epidemiology, and biochemistry of amatoxins and phallotoxins. Five distinct species of this section have been identified in Europe to date: *A. phalloides*, *A. virosa*, *A. verna*, the recently introduced North American species *A. amerivirosa*, and *A. vidua* sp. nov., which is a new name proposed for the KOH-negative Mediterranean species previously described as *A. verna* or *A. decipiens* by various authors. Epitypes or neotypes are selected for species lacking suitable reference collections, namely *A. verna* and *A. virosa*. Three additional taxa, *Amanita decipiens*, *A. porrinensis,* and *A. virosa* var. *levipes* are here considered later heterotypic synonyms of *A. verna*, *A. phalloides,* and *A. amerivirosa*, respectively.

## 1. Introduction

Lethal species of the genus *Amanita* Pers. (*Amanitaceae*, *Agaricales*, *Agaricomycetes*, *Basidiomycota*, *Fungi*), including the white-coloured taxa in sect. *Phalloideae*, which are commonly called ‘destroying angels’, are among the most well-documented causes of intoxication by fungi, with reports dating back to ancient history. Despite the information campaigns periodically launched by public health services and mycological societies, intoxications caused by these species are still reported every year, resulting in the most severe cases of mushroom poisoning in Europe and North America [1,2,3]. The first lethal species to be formally named were *A. phalloides* (Vaill. ex Fr.) Link (basionym *Agaricus phalloides* Vaill. ex Fr.) and *A. verna* Bull. ex. Lam. (protonym *Ag. bulbosus vernus* Bull.), both found near Paris, France [4,5], and *A. virosa* Bertill. (basionym *Ag. virosus* Fr.) from Sweden [6]. Amatoxin-producing species, in most cases identified as *Amanita phalloides* (listed here among the ‘destroying angels’ because of its whitish forms: see below), are the most frequent cause of lethal intoxication by fungi in the USA [1], China [7], Bulgaria [8], Catalonia [9], Czech Republic [10], France [2,11], Northern Italy [12,13], Germany [14], South Korea [15], Switzerland [16], and Turkey [17].

However, poisonings with amatoxins in Europe are also caused by other taxa, such as *A. verna*, *A. virosa*, as well as multiple species of *Lepiota* [18], and the lineages around *Galerina marginata* (Batsch) Kühner [19,20,21,22,23]. Many lethal intoxication cases have been reported from Eastern and Central Asia, attributed either to *A. verna* [24,25] or *A. virosa* [26], and intoxications in spring seem to be even more frequent than those occurring in autumn in Iran [27]. In Lebanon, several spring intoxications were reported among Syrian migrants after the consumption of white phalloid-like *Amanita* (Bou Dagher-Kharrat, unpublished data). This was also the case in Turkey and Poland among Syrian and Afghan refugees, respectively [28,29]. A critical approach to treat these cases is to disentangle recurrent taxonomic controversies regarding the identity of *A. verna* and *A. virosa*, as well as to characterize the toxicological profiles of related taxa. For example, as noticed by [25], the amount of amatoxins in *A. verna* is surprisingly variable in the literature, leading Preston et al. [30] to hypothesize the existence of several species or varieties.

Recent phylogenetic accounts on sect. *Phalloideae* [31,32,33,34,35,36,37] have focused primarily on American, Asian, and African species, with very few European data included. According to Neville & Poumarat [38], the only species of sect. *Phalloideae* occurring in Europe were *A. phalloides*, *A. verna*, *A. decipiens* (Trimbach) Jacquet. (basionym *A. verna* var. *decipiens* Trimbach), *A. dunensis* Bon & Andary, and *A. porrinensis* Freire & M.L. Castro. In addition, a number of infraspecific taxa of these species were described: *Amanita phalloides* var. *alba* Costantin & L.M. Dufour, *A. phalloides* f. *citrina* J.E. Lange, *A. phalloides* var. *euphalloides* Maire, *A. phalloides* var. *larroquei* F. Massart & Beauvais ex F. Massart, *A. phalloides* var. *moravecii* Pilát, *A. phalloides* var. *pulla* Killerm., *A. phalloides* var. *striatula* Peck, *A. phalloides* f. *umbrina* Ferry, *A. verna* f. *ellipticospora* E.-J. Gilbert, *A. verna* var. *grisea* Massee, *A. verna* var. *tarda* Trimbach, *A. virosa* var. *aculeata* Voglino, *A. virosa* var. *levipes* Neville & Poumarat. The taxonomic status of most of these taxa has not yet been confirmed with genetic tools.

The aim of the present work is to address existing doubts about the taxonomic status of the most relevant species and varieties of *Amanita* sect. *Phalloideae* occurring in Europe and western Asia, by providing an accurate genetic profile of the type material existing in official herbaria or by designating new types selected among modern collections. The content in amatoxins and phallotoxins of all species analyzed was also chemically characterized, both qualitatively and quantitatively. Finally, an epidemiological analysis of data obtained from the Spanish health system was carried out to assess the risk of phalloidian poisonings in spring and fall in the different regions of this Mediterranean country.

## 2. Materials and Methods

### 2.1. Herbarium Collections and Morphological Studies

Basidiomata of white species of *Amanita* sect. *Phalloideae* were collected in France, Italy, Lebanon, Spain, and Sweden by various collectors. Samples to be studied were selected amongst available collections based on their conservation state, collection date, and additional documentation available (collection data, images). Samples were deposited in the herbaria of the Universidad de Alcalá (AH), Conservatoire et Jardin botaniques de la Ville de Genève (G), Université de Lille (LIP), Centro de Investigaciones Forestales de Lourizan (LOU), Real Jardín Botánico de Madrid (MA), Muséum National d’Histoire Naturelle de Paris (PC), Muséum d’Histoire Naturelle de Nice (NICE). Herbarium codes follow Thiers (continuously updated).

Microscopical observations of rehydrated herbarium specimens were conducted at the University of Lille by P.-A. Moreau and the University of Alcalá by G. Moreno, on hand-made longitudinal sections of lamellae, as well as cross-sections and tangential sections of pileipellis, universal veil, and partial veil in a 5% potassium hydroxide aqueous solution (5% KOH). Basidiospores naturally deposited on the surface of the partial veil or the apex of the stipe were observed in 5% KOH and Melzer’s reagent. Microscopic pictures were taken with a Nachet Andromède microscope equipped with a Motic 3.0 digital camera; spore dimensions were estimated with the software Piximètre 5.10 (A. Henriot, http://ach.log.free.fr/Piximetre accessed on 1 March 2022, continuously updated). Spore measurements include extreme measured values (in brackets) and 1st–9th deciles; Q represents the spore ratio (length/width), cited by 1st–9th deciles, and its mean value Q_av_. The shape of spores is interpreted following Bas [39]: subglobose means Q = [1.05–1.15], broadly ellipsoid [1.15–1.3], ellipsoid [1.3–1.6], and cylindrical [2.0–3.0].

### 2.2. Phylogenetic Studies

Total DNA was extracted from herbarium specimens using a standard method based on cetyltrimethylammonium bromide (CTAB) [40], or with the REDExtract-N-Amp^tm^ Plant PCR Kit (Sigma-Aldrich, St. Louis, MO, USA), following the manufacturer’s instructions. Polymerase chain reactions (PCRs) [41] included 35 cycles with an annealing temperature of 54 °C. Primers ITS1F and ITS4 [42,43] were employed to amplify the internal transcribed spacer nuc-rDNA region (ITS), while LR0R, LR1, and LR7 [44,45,46] were used for the 28S nuc-rDNA large ribosomal subunit (LSU). PCR products were checked in 1% agarose gels, and positive reactions were sequenced with one or both PCR primers (depending on the quality of the resulting sequences). Chromatograms were edited by hand to correct reads at heteromorphic sites and other putative errors using MEGA5 [47] or Codon Code Aligner 4.1.1 (CodonCode Corp., Centerville, MA, USA).

The ITS and LSU sequences obtained were aligned with the closest matches from BLAST searches in the public databases. The sequences retrieved (Appendix A) came mainly from [48,49,50,51,52], among others. The resulting dataset was aligned in MEGA5 software using its ClustalW application followed by manual correction. The concatenated loci (677 nt ITS, 554 nt LSU) were subjected to Gblocks [53], which removed 221 ambiguously aligned positions (165 nt ITS, 56 nt LSU), to obtain a final alignment with 512 nt from ITS (301 nt variable), and 498 nt from LSU (139 nt variable). Each region was loaded in PAUP* 4.0b10 [54] and subjected to MrModeltest 2.3 [55] to infer the best-fitting evolutionary model. MrBayes 3.2.6 [56] was employed to conduct a Bayesian inference (BI) analysis using model GTR + G+I (nst = 6, rates = invgamma, statefreqpr = dirichlet) in each partition (ITS and LSU), two simultaneous runs, four chains, temperature set to 0.2, and sampling every 100th generation. Finally, a full search for the best-scoring maximum likelihood (ML) tree was performed in RAxML 8.2.10 [57] using the standard search algorithm with the GTRCAT approximation as recommended by the manual for data sets >50 taxa, data partitioned as for Bayesian analysis, and 2000 bootstrap replications. All analyses were run locally. The significance threshold was set above 0.95 posterior probability (PP) for BI and above 70% bootstrap proportion (BP) for ML.

### 2.3. Toxicological Studies

High performance liquid chromatography (HPLC) was employed to quantify the amatoxins present in selected samples from the species inferred by phylogenetic analyses (Appendix A). All extractions were repeated three times. Since the pileus of poisonous species of *Amanita* is known to contain large amounts of toxins [58,59], a portion of the cap (50 mg of dry material) was taken from each sample and finely ground. One ml of a mixture of MeOH/H*_2_*O/HCl (5:4:1, *v*/*v*/*v*) was added as previously described [25,59,60,61,62], sonicated for 15 min, and then filtered and centrifuged. Three different fruitbodies of PAM19050401 (harvested in the same location and time, named a, b, and c) were analyzed, each one being extracted three times as the other samples.

Chromatographic separation and detection for quantitative analyses were performed on an Ultimate 3000 instrument that included a quaternary pump, a degasser, an automatic sampler, and a DAD detector (Thermo Fisher Scientific Inc., San Jose, CA, USA). The system was operated using the Chromeleon software, version 2.0. Chromatographic separation was achieved on an ODS Hypersyl C18 column (250 × 4.6 mm, 5 μm, Thermo Fisher Scientific Inc., San Jose, CA, USA), with a column temperature maintained at 35 °C. Analyses were performed at a flow rate of 1 mL/min, using solvent A (0.02 M ammonium acetate/acetonitrile 90:10 *v*/*v*) and solvent B (0.02 M ammonium acetate/acetonitrile 76:24 *v*/*v*) as previously described [60,61,63]. HPLC-grade acetonitrile (Carlo Erba Reagents, Val de Reuil, France) and water (PanReac Quimica SLU, Castellar del Valles, Spain) were employed. The gradient profile was set up as follows: 100% A for 4 min, then 57% B for 16 min, then 100% B for 10 min, and finally 100% A.

Standards of α-amanitin, β-amanitin, phallacidin, and phalloidin were purchased from Sigma-Aldrich (Saint-Louis, MO, USA) and stocked at −20 °C. The toxins were freshly prepared and mixed together to obtain calibration solutions at 1.95, 3.91, 7.81, 15.63, 31.25, 62.5, 125, 250, and 500 µg/mL in MeOH/H_2_O (50:50). Individual calibration curves were produced for each toxin and shown to be linear over the range of interest (R2 > 0.9995, *n* = 3, Appendix A). The UV/Vis spectra were recorded in the 200–400 nm range and chromatograms were acquired at 254 nm (unspecific wavelength), 295 nm (λ_max_ for phallacidin and phalloidin), and 305 nm (λ_max_ for α-amanitin and β-amanitin). Extracts obtained from the different species of *Amanita* studied in the present work were analyzed directly without any prior dilution. For each peak, a UV spectrum was obtained and compared with those of commercial standards to confirm their identification. Contents of amatoxins and phallotoxins were expressed in mg of toxin/g of dried cap.

### 2.4. Epidemiological Studies

The epidemiological analysis focused on phalloidian intoxications in Spain, targeting two main indicators: (1) cumulative incidence of poisoning according to time of year and (2) geographical and seasonal distribution of poisoning incidence. Primary data were gathered from the following sources:Ministry of Science and Innovation–National Center of Epidemiology (CNE)–Sistema de Vigilancia de la Red Nacional de Vigilancia Epidemiológica (RENAVE), SiViEs platform: for the period 1980–2020, outbreaks in which the agent was a fungal toxin and those associated with another agent in which mushrooms were consumed. In this database, only clusters (*n* ≥ 2 patients) are registered. To target putative phalloidian cases, the criterion “Incubation time” was selected for values > 6 h (when recorded) coupled with “hospitalization”, assuming that phalloidian poisonings required hospitalizations. Cases involving non-phalloidian fungal species were discarded.Ministry of Health, Consumer Affairs and Social Welfare-Specialized Health Care Activity Register (CMBD and RAE-CMB data). These databases record individual hospital exits and summarize care pathways (diagnostics, hospitalization times, etc.). For the period 1997–2015 (MBDS-H; ICD 9), clinical criteria 988.1 (toxic effect of mushrooms and mushrooms) and 573.1 (hepatitis unspecified) were selected. For the period 2016–2020 (RAE-CMD; IC10), clinical criteria T62.0X1A (toxic effect of ingested mushrooms, unintentional/accidental) and K71.2 (toxic hepatopathy with acute hepatitis) were retained.

## 3. Results

### 3.1. Phylogeny

The Bayesian inference of phylogeny based on ITS and LSU data (Figure 1) is consistent with previously published organization of *Amanita* sect. *Phalloideae* [31,33,34,35,36,37,64], although the limited signal provided by the ribosomal regions alone does not resolve most supraspecific nodes supported by the analyses, including protein-coding genes. The Mediterranean ‘destroying angels’ include at least seven different species. Five of them cluster inside sect. *Phalloideae*: (1) *Amanita phalloides* (= *A. phalloides* var. *alba*, = *A. porrinensis*), (2) *A. verna* KOH+, (3) *Amanita verna* KOH-, (4) *A. virosa,* and (5) *A. amerivirosa* Tulloss, L.V. Kudzma & M. Tulloss (= *A. virosa* var. *levipes*). Another species (6) is nested within sect. *Strobiliformes* (*Amanita strobiliformis*), and the last one (7) belongs in sect. *Validae* (*A. phalloides* var. *larroquei* = *A. citrina*). *Amanita phalloides* is significantly related to *A. subjunquillea* S. Imai, which includes a white variety, *A. subjunquillea* var. *alba* Zhu L. Yang. While both species did not receive reciprocal significant support in the present analysis, they are considered independent taxa until a more complete analysis is done to test their actual status. *Amanita virosa* and *A. amerivirosa*/*A. virosa* var. *levipes* are significantly related to *A. subpallidorosea* Hai J. Li, but their phylogenetic relation with *A. ocreata* Peck, found in the multigenic analysis performed by Codjia et al. [34] could not be recovered with the present analysis. All Mediterranean species in sect. *Phalloideae* are nested within a ‘Northern Hemisphere lineage’, significantly distinct from the remaining taxa, which in previous works [34,35,36,37], constitute three different monophyletic lineages. ITS rDNA sequences of *A. strobiliformis* are 92% similar to those of *A. sabulicola* S. Morini et al., which is by now the closest relative [65]. In turn, the ITS sequence of *A. phalloides* var. *larroquei* (specimen provided by its author F. Massart) is almost 98% similar to sequences of *A. citrina* Pers. (i.e., MH508311), and therefore probably belongs to this species. The taxonomy of these species is updated below according to these results.

### 3.2. Taxonomy

1.***Amanita phalloides*** (Vaill. ex Fr.) Link, Handb. Erk. Gew. 3: 272 (1833) Figure 2E, Figure 3E and Figure 4E
≡*Agaricus phalloides* Vaill. ex Fr., Syst. mycol. (Lundae) 1: 13 (1821), *nom. sanct.* [protonym *Fungus phalloides* Vaill., Bot. paris. (Paris): 74, Table 14, Figure 5 (1723), inval.]; *Venenarius phalloides* (Vaill. ex Fr.) Murrill, Mycologia 4(5): 240 (1912); *Amanitina phalloides* (Vaill. ex Fr.) E.-J. Gilbert, in Bresadola, Iconogr. mycol., Suppl. I (Milan) 27: 78 (1940)
=*Agaricus insidiosus* Letell., Annls Sci. Nat., Bot., sér. 2 4: 35 (1835); *Amanitopsis insidiosa* (Letell.) Sacc., Syll. Fung. 9: 2 (1891); *Amanita insidiosa* (Letell.) Bigeard & Guillemin, Fl. champ. sup. France Edn 2: 14 (1913); *Amanita phalloides* f. *insidiosa* (Letell.) E.-J. Gilbert, Le genre *Amanita*: 41 (1918)=*Amanita phalloides* var. *alba* Costantin & L.M. Dufour, Nouv. Fl. Champ., Edn 2 (Paris): 256 (1895)=*Amanita phalloides* f. *citrina* J.E. Lange, Dansk bot. Ark. 2(3): 8 (1915)=*Amanita phalloides* var. *pulla* Killerm., Denkschr. Bayer. Botan. Ges. in Regensb. 18: 4 (1930)=*Amanita phalloides* var. *euphalloides* Maire, Mém. Soc. Sci. Nat. Maroc. 45: 103 (1937)=*Amanita phalloides* var. *moravecii* Pilát, Česká Mykol. 20(1): 25 (1966)=*Amanita dunensis* R. Heim ex Bon & Andary, Docums Mycol. 13(50): 13 (1983); *Amanita phalloides* f. *dunensis* R. Heim, Revue Mycol., Paris 28: 9 (1963) [nom. inval., Art. 36.1] (synonymy proposed here)=*Amanita verna* var. *tarda* Trimbach, Annales du Muséum d’Histoire Naturelle de Nice 91(3): 85 (1999); *Amanita tarda* (Trimbach) Contu, Boll. Gruppo Micol. ‘G. Bresadola’ (Trento) 43(2): 83 (2000)



Invalid names
=*Hypophyllum virosum* Paulet Traité champ. Comest. (Paris): 2: 326 (1793) [inval., op. rejic.]=*Amanita viridis* Pers., Tent. disp. meth. fung. (Lipsiae): 67 (1797) [illegit., non *A. viridis* Huds.)]=*Agaricus virosus* Vittad., Descr. fung. mang. Italia: 135 (1835) [illegit., non *A. virosus* Sow.]=*Amanita andaryi* Mornand, Docums Mycol. 22(88): 12 (1993) [inval., introduced as nom. nov. based on “*A. virosus* var. *albus* Vittad. 1835: 143”, but this name does not exist in the cited book]

Pileus subglobose at first, then convex-flattened, lacking an umbo, 6–15 cm in diam.; surface easily detachable, smooth, shiny, thin, frequently greenish with a darker centre, but also yellowish-greenish, pale with greenish tinges, or rarely pure white (*A. phalloides* var. *alba*); with adnate radial fibrils, olivaceous to dark olivaceous in color; margin not striated, incurved when young, flat when mature. Lamellae very crowded, free, with pure-white or sometimes greenish lamellulae; edge white, but slightly yellowish or greenish near the pileal margin. Stipe 7–18 × 1–2 cm, cylindrical to subcylindrical, white with greenish tinges; with a white, sack-like, membranous volva, detachable, formed by 3–4 lobes, free in their upper half. Partial veil an annulus hanging high on the stipe, longitudinally striate, membranous, thick, white with citrine tinges. Universal veil without remnants on pileus surface. Flesh white, with greenish or yellowish tinges under the cuticle; odourless or with a slightly pleasant smell (resembling that of roses) when young, urine-like and disagreeable when mature; taste sweetish. Reactions to NaOH or KOH nil to faintly yellowish on white specimens.

Spores [3/72] mostly broadly ellipsoid (60%), also subglobose (30%) or ellipsoid (20%), (7.5–) 8.0–9.5 (–10.0) × 6.7–7.8 (–8.5) µm, Q = 1.1–1.3, Q_av_ = 1.2, smooth, hyaline and amyloid. Basidia 4-spored, 38–52 × 9–12 µm, clampless. Pileipellis an ixocutis formed by gelified filamentous hyphae 1.5–5 µm in diam., clampless. Partial veil formed by cylindrical hyphae 1.5–5 µm in diam., clampless. Volva filamentous, formed by cylindrical hyphae 2–5.5 µm in diam., clampless, with rare sphaerocytes.

Habitat and distribution: very frequent, isolated, or scattered basidiomes can be found in autumn, forming mycorrhizal associations with *Quercus* as well as with other broadleaved trees, such as *Castanea*, *Corylus*, *Betula* or *Fagus*, less commonly with conifers (*Cedrus*, *Pinus*). Found in temperate, Mediterranean, and boreal habitats.

Examined material: FRANCE: Haute-Corse, Ghisonaccia, Forêt de Piniu, under *Pinus pinaster* in fixed sand dune, 5 November 2019, leg. P.-A. Moreau, PAM19110516, 0401670 (LIP). Oise, near Saint-Sauveur, forêt de Compiègne, broadleaved forest, 22 August 1960, leg. H. Romagnesi, HR60.109 (PC, as ‘*Amanita verna*’). SPAIN: Cádiz, Jimena de la Frontera, Parque Natural de los Alcornocales, under *Quercus suber* and *Q. canariensis*, 29 November 2003, leg. F. Prieto & A. González, 34,748 (AH). Madrid, between Bustarviejo and Pto. de Canencia, under *Quercus pyrenaica*, 1 October 1975, leg. G. Moreno, 94 (AH).

Notes: *Amanita phalloides* is a very common autumnal species described in detail in many publications and web sites, i.e., [38,67,68]. Freire & Castro [69] did not recognize *A. phalloides* in the odd-shaped specimen described by them (and validated by Castro [70]) as *A. porrinensis*; these two species cannot be separated from one another with the current sequencing results, and so they are considered conspecific. Therefore, we propose to downgrade the unusual Mediterranean taxon observed by Freire & Castro [69] and redescribed by Neville et al. [71] to a form of *A. phalloides*.

2.***Amanita phalloides* f. *porrinensis*** (Freire & M.L. Castro) G. Moreno & Olariaga, comb. nov. Figure 2G and Figure 3E
MycoBank number: MB 842794
≡*Amanita porrinensis* Freire & M.L. Castro, Mykes 1: 59 (1998) [also An. Jard. bot. Madr. 44(2): 533 (1987), inval., art. 39.1]

Examined material: ITALY: Ferrara, Comacchio, Bosco di Volano, forest with *Pinus pinaster* and *P. pinea*, 17 October 1998, leg. G. Monterumici, P. Neville 00111702 (G). Sardegna, Seneghe, forest of *Quercus suber* and *Q. ilex*, 1 November 2020, leg. S. Nioi, ALV30282, 0402258 (LIP). SPAIN: Pontevedra, Vigo, A Madroa, mixed forest with *Quercus robur* and *Pinus pinaster*, 28 October 1984, leg. J. Diz (G. M. ‘O Porriño’), Fungi-3670 (LOU, holotype). Idem, 7 November 1991, leg. L. Freire, M. Castro & J. Diz, Fungi-27321 (MA).

Pileus campanulate, with a wide and obtuse central umbo, 5–7.5 cm in diam.; surface pure white, not changing, smooth, not striate nor fibrillose, lacking any remnants of the universal veil, difficult to separate from the context underneath; margin not striated, inrolled at first, then flat when mature. Lamellae very crowded free, with pure white lamellulae; edge white; spore print white. Stipe 10–20 × 2–3 cm, cylindrical, sinuose, enlarged towards the base, pure white, sericeous to slightly floccose; annulus white, hanging high on the stipe, membranous, thin, soon disappearing; universal veil forming a fragile sack-like membranous volva, adhering, detachable, without remnants on the pileus surface. Flesh white, not changing, odourless; taste slightly sweet. Reactions: pileipellis turning citrine yellow with KOH 10% according to Freire & Castro [69], although the same authors report a weak pale yellowish reaction in the herbarium notes. Neville et al. [71] report an intensely yellow reaction on lamellae, weaker in the pileus and stipe; positive to Wieland test.

Spores broadly ellipsoid to ovoid or rarely subglobose, 7–10 × 6–7.5 µm, smooth, hyaline and amyloid. Basidia 4-spored, 35–40 × 10–11 µm, clavate, clampless. Pileipellis an ixocutis formed by filamentous hyphae 2–6 µm in diam., clampless, slightly gelified. Annulus formed by cylindrical hyphae 2.5–8 µm in diam., lacking sphaerocytes and clampless. Volva filamentous, formed by cylindrical hyphae 2–5.5 µm in diam., clampless, without sphaerocytes, but sometimes with clavate elements.

Notes: a very rare albino form of *A. phalloides* with a characteristic obtuse wide umbo and squamose or sericeous stipe. Only a few collections are known, found in autumn, in Spain [69], as well as Italy and Switzerland [71,72,73]. A toxic species according to the results of the Wieland test performed by the original authors [69]. The present results based on ribosomal data suggest that *A. porrinensis* is conspecific with *A. phalloides*. Sequences from several protein-coding genes (tef1, rpb2) obtained from a recent collection from Italy (ALV30282) do not show differences with those obtained from *A. phalloides* available in public databases (data not shown). *Amanita porrinensis* is here downgraded to a form of *A. phalloides* to warn about this unusual, but probably lethal, morphological variant.

3.***Amanita amerivirosa*** Tulloss, L.V. Kudzma & M. Tulloss, Amanitaceae 1(4): 5 (2021) Figure 2A–C, Figure 3A, Figure 4C and Figure 5
=*Amanita virosa* var. *levipes* Neville & Poumarat, Fungi europ. 9: 600 (2004)


Examined material: FRANCE: Loire-Atlantique, Nantes, under *Quercus petraea*, November 2019, leg. R. Chéreau, PAM19110001, 0401671 (LIP). Idem, under *Quercus* sp., November 2019, leg. G. Ouvrard, PAM19110002, 0401672 (LIP). Pas-de-Calais, Baincthun, forêt domaniale de Boulogne, under *Quercus petraea* and *Carpinus betulus* on a wet clay-calcareous soil, 24 July 2021, A. Gasch & P.-A. Moreau, PAM21072401, 0402251 (LIP). Val-d’Oise, Montmorency, forêt domaniale, under *Quercus* spp. and other broadleaved trees, 14 June 2019, leg. N. Journe FR2019667, 0402250 (LIP). Mayenne, Forêt communale d’Ombrée D’Anjou-Combrée, acidophilic atlantic *Quercus petraea* forest, 11 August 2021, A. Gasch & P.-A. Moreau, PAM21081110, 0402252 (LIP).

Pileus 4–15 cm in diam., originally conical-campanulate, expanding to convex-flat with a persistently convex center, never depressed; surface pure white and remaining so or sometimes becoming light ochraceous-pinkish at the center with age, greasy to nearly dry, shiny-silky when dry, smooth; margin incurved then expanded and flexuose when mature, smooth, never striated. Lamellae crowded, 80–100, reaching the stipe, typically with one lamellula per lamella, smooth to nearly smooth, cream white, with slight pinkish tones with age; edge white, floccose. Stipe 9–20 (–25) × 0.8–2.5 cm, cylindrical throughout or progressively thinner towards the apex, bulbous-sphaerical at the base (which can be up to 3.5 cm thick), pure white, slightly floccose towards the apex, below the annulus subtly silky-zebrate, becoming glabrous with age; partial veil a membranous, persistent annulus hanging 2–3 cm down the apex, rarely laciniate and then hanging at the margin, not adhering to the edges; universal veil forming an ample membranous volva, rarely dissociated on the pileus, white, with pale pinkish-ochraceous patches on outer surface. Context white, slightly foxy-spotted on damaged surfaces, light lemon yellow when dry; smell distinctly iodine-like in the bulb, elsewhere not distinct, fungoid by alteration; taste mild, fungoid. Reaction to 5% KOH chrome yellow on all surfaces and context. Exsiccates turn entirely light lemon yellow after a few days. Spore print pure white.

Spores [2/40] (7.1–) 8.1–10.2 (–11.8) × (6.7–) 7.3–9.3 (–11.1) µm, Q = 1.00–1.20, Q_av_ = 1.1, subglobose (60–80%) to broadly ellipsoid (20–40%), smooth, amyloid, with dense yellow content in KOH at maturity, rather thick-walled and not easily collapsing. Hymenophoral trama a 50–60 µm thick mediostratum, reaching 80 µm towards the pileus, made of broad physaloid hyphae 8–25 µm wide, mixed with slender hyphae 4–8 µm wide, divergent towards the hymenium. Subhymenium 10–15 µm thick, ramose, faintly developed. Basidia 35–43 × 9–12 µm, mostly 4-spored with a significant number of 2-spored basidia. Lamellae edge sterile, terminal inflated cells measuring 18–45 × 12–18 µm, shortly clavate to sphaeropedunculate, wall yellowish up to 0.5 µm thick, often incrusted by yellowish mucoid deposits. Pileipellis scarcely differentiated, 30–50 µm thick, an ixotutis of slender hyphae 1.5–4 µm wide, with yellow wall irregularly thickened, forming internal and external yellow concretions; subpellis not distinct. Universal veil 2-layered; outer layer made of yellowish hyphae 3–12 µm wide, with yellow wall covered by mucoid deposits; inner layer made of colourless hyphae 2.5–14 µm wide, with smooth wall up to 0.5 µm thick, mostly long and fasciculate. Partial veil on the upper surface made of intricate filamentous hyphae with numerous globose, cylindrical elements with yellow, strongly granulose surface, 3–12 µm wide. Lower surface with numerous globose, cylindrical to elongate elements 20–60 × 7–25 µm.

Notes: *Amanita amerivirosa* is a very recently described taxon [37], so far only documented by DNA sequences from NE America where it has been recorded for a long time under the name *A. virosa*. We confirm here Neville & Poumarat’s [38] hypothesis that *Amanita virosa* var. *levipes* described from the French Atlantic coast is the same fungus, likely not indigenous in Europe.

4.***Amanita verna*** Bull. ex Lam., Encycl. Méth. Bot. (Paris) 1(1): 113 (1783) Figure 2F,I–M, Figure 3C, Figure 4F,G and Figure 6

Protonym: *Agaricus bulbosus vernus* Bull., Herb. Fr. (Paris) 3: Table 108 (1782–1783) [inval., Art. 32.1(c)]; Agaricus virosus [unranked] vernus (Bull. ex Lam.) Fr., Epicr. syst. mycol. (Uppsala): 4 (1838); Amanita phalloides [unranked] verna (Bull. ex Lam.) Fr., Hymenom. Europ. (Uppsala): 18 (1874); Amanita phalloides var. verna (Bull. ex Lam.) Lanzi, Bull. Soc. Hist. nat. Afr. N. 7(6): 145 (1916); Amanitina verna (Bull. ex Lam.) E.-J. Gilbert, in Bresadola, Iconogr. mycol., Suppl. I (Milan) 27: 78 (1940); Venenarius vernus (Bull. ex Lam.) Murrill, Lloydia 11: 104 (1948)

Typification: lectotype (designated here): Bulliard, Herb. Fr. 2: Table 108 (1782–1783), MycoBank MBT 10005511. Epitype (designated here): FRANCE: Oise, Apremont, 19 Jul 1956, leg. H. Romagnesi, HR56.31 (PC), as a support to the lectotype designated above, MycoBank MBT 10005512.
=*Agaricus vernalis* Bolton, Hist. fung. Halifax (Huddersfield) 2: 48, Table XLVIII (1788)=*Amanita verna* var. *ochroleuca* Forq. ex Quél., 1888, Fl. mycol. Fr.: 309; *Amanita phalloides* var. *ochroleuca* (Forq. ex Quél.) Quél. & Bataille, 1902, Fl. Amanites et lépiotes: 32; *Amanita ochroleuca* (Forq. ex Quél.) Bigeard & Guillemin, 1913, Fl. champ. sup. France 2nd ed.: 14=*Amanita verna* var. *decipiens* Trimbach, Riviéra Scientifique, 1970(1): 18 (1970); *Amanita decipiens* (Trimbach) Jacquet., Docums Mycol. 22(86): 30 (1992)

Invalid names
=*Amanita verna* f. *ellipticospora* E.-J. Gilbert, in Bresadola, Iconogr. Mycol., Suppl. II (Milan) 27: 320 (1941) [inval., art. 38.1 and art. 39.1]

Pileus 3–9 cm diam., globose at first then convex to flat, soon flattened to somewhat depressed at center; surface smooth, slightly greasy when moist, pure white at first, somewhat ochraceous towards disk with age; margin convex, only later expanding, never striate. Lamellae crowded, 70–80 reaching the stipe, 1 (-2) lamellulae per lamella, free to somewhat adnexed, pure white when young, often with a subtle pinkish tone when ageing; edge white, thin, smooth to somewhat serrulate. Stipe 6–15 × 0.8–1.5 cm, equal to slightly narrowed at apex, abruptly bulbous-sphaerical at base, not or only faintly striate at apex, smooth, silky to subtly floccose below annulus; partial veil a membranous, persistent but thin annulus usually hanging high on the stipe, white, slightly striate on the upper side, smooth; universal veil a membranous, rather thin volva, white to somewhat ochraceous on surface. Flesh white, unchanging; smell none, somewhat fungoid on adults; taste mild. Reaction chrome-yellow to 5% KOH on all surfaces and context, except inner layer of volva.

Spores [3/60] usually mostly broadly ellipsoid (50%) and partly subglobose and ellipsoid (20–30% each), some lacrymoid or pyriform in side view, (7.4–) 8.0–11.1 (–11.3) × (6.1–) 6.4–9.1 (–9.7) µm, Q = 1.15–1.50, Q_av_= 1.2, smooth, amyloid. Basidia 4-spored, 42–58 × 7.5–10 µm, clavate with tapering base, with yellow granular content at maturity, then filled with yellow, amorphous to crystalloid necropigment. Hymenophoral trama made up of a mediostratum 30–50 µm wide (on exsiccate), with parallel slender hyphae, more divergent towards subhymenium, 2–12 µm wide. Subhymenium 30–45 µm thick, with polygonal, lobate to cordiform elements, 12–24 × 7–18 µm. Lamellae edge sterile to substerile, consisting of clusters of terminals inflated, sphaeropedunculate cells measuring 10–25 × 8–15 µm, with yellowish wall up to 1.2 µm thick, with abundant yellow granulations on surface. Pileipellis 100–120 µm thick, with a suprapellis 30–40 µm thick made of slender hyphae 3–4 µm wide, with abundant yellowish mucoid deposits in KOH. Subpellis not differentiated. Context of the pileus made up of thick, intricate hyphae up to 10–12 µm broad. Partial veil made up of intricate cylindrical hyphae 3–8 µm wide, with scattered strongly granular content; upper side with fascicles of globose to piriform elements measuring 15–35 × 10–14 (–20) µm, mostly with thickened walls; outer surface made up of more or less parallel hyphae. Universal veil 1-layered, made up of both slender and thick hyphae 3–18 µm wide, intricate, smooth, with interspersed yellow granulations.

Ecology and distribution: ectomycorrhizal with *Fagaceae*, especially *Quercus ilex*, *Q. pubescens*, *Q. pyrenaica*, *Q. robur* and *Q. suber*, and also *Pinus pinea*, exceptionally with *Castanea sativa* or *Fagus sylvatica*, on acidic to moderately basic soils, known from thermophilous regions in Western Europe and North Africa, probably widespread across the Mediterranean Basin; mostly fruiting in spring (March to June), occasionally observed in late autumn.

Examined material: FRANCE: Oise, Apremont, in the humus of a broadleaved forest, 19 July 1956, leg. H. Romagnesi, HR56.31 (PC). Oise, Coye-la-Forêt, near Poteau du Crochet, broadleaved forest, 20 June 1971, leg. H. Romagnesi, HR71.42 (PC). Alpes-Maritimes, Berre-les-Alpes, June 1969, J. Trimbach, 2010.2.148 (NICE, holotype of *A. verna* var. *decipiens*). Landes, Bias, under *Quercus pubescens* on sandy soil, J.-C. Déiana, J. Guinberteau & P.-A. Moreau, PAM01042903, 0402249 (LIP, not sequenced). ITALY: Lazio, Anguillara Sabazia, among litter in acidic soil, along a sidetrack on a steep slope facing north in a pure stand of *Quercus ilex*, 185 m a.s.l., ca 100 m from the shore of lake Bracciano, 16 May 2021, leg. M. Gelardi, F. Costanzo & O. Gelardi, MG846-BIS, 0402254 (LIP). Lazio, Bracciano, Bosco di S. Celso, among litter in acidic soil in a pure stand of *Quercus cerris*, 190 m a.s.l., ca 200 m from the shore of lake Bracciano, 1 May 2021, leg. A. Appolloni & M. Tosoni, MG847, 0402255 (LIP). Tuscany, Pisa, Vecchiano, San Piero a Grado, Tenuta di Salviati, in a forest of *Pinus pinea* and *Quercus ilex*, 17 May 2019, leg. M. Raumi, Raumi ‘Amanita Salviati’ ALV21204, 0002275 (LIP). SPAIN: Ávila, Casillas, under *Castanea sativa*, leg. L. Rubio-Casas, 25 May 2008, 31,839 (AH). Ávila, El Tiemblo, under *Quercus pyrenaica*, leg. L. Rubio-Casas, J.C. Campos, L. Rubio-Roldán & J. Hernanz, 14 June 2008, 31,848 (AH). Idem, 31,849 (AH). Cáceres, Miramontes, Cerro la Garrapata, under *Quercus pyrenaica* and *Cistus ladanifer*, leg. G. Moreno & E. Arrojo, 6 May 2000, 19,695 (AH). Madrid, leg. Soc. Micol. Madrid, 5 May 2002, 31,985 (AH). Idem. 31,986 (AH). Madrid, El Escorial, under *Quercus pyrenaica*, leg. B. Nilsson, 6 June 2008, 31,843 (AH). Madrid, El Pardo, under *Quercus ilex* subsp. *ballota*, leg. V. Sánchez, 2 June 2008, 31,842 (AH). Madrid, Villa del Prado, under *Quercus ilex* subsp. *ballota*, leg. J.C. Campos, 25 April 2003, 37,755 (AH). Idem, leg. J. Hernanz, J.C. Campos & J. Campos, 1 May 2004, 37,726 (AH). Salamanca, under *Quercus* spp., June 2019, leg. P. García Jiménez, PAM19060001, 0402244 (LIP). Toledo, Hinojosa de San Vicente, leg. L. Rubio-Casas, 22 May 2004, 31,987 (AH). Toledo, Hormigos, under *Quercus ilex* subsp. *ballota*, leg. J.C. Campos & J. Hernanz, 1 May 2004, 31,991 (AH). Idem, 37,725 (AH). Idem, leg. A. Zapata, 8 May 2004, 37,729 (AH).

Notes: this lethal species can be easily confused with immature specimens of *Agaricus*. Unexpanded young basidiomata can also be confused with the edible species *Amanita ponderosa* Malençon & R. Heim (‘gurumelo’), a highly regarded mushroom in south-eastern Spain (Huelva, Badajoz, Cáceres; [74]). The results obtained in the present work (Figure 1) show that KOH+ and KOH- specimens of *A. verna* actually belong to distinct species. All KOH- specimens examined were found in Mediterranean habitats of southern Europe. Romagnesi’s collection HR83.65 at PC herbarium (supposed *A. verna* KOH- from northern France; [75]) was actually made of two specimens, one with white pileus, the second with distinct greenish shades. Sequence data obtained from them revealed that both are actually specimens of *A. strobiliformis* with an atypical membranous volva.

If the original concept of *A. verna* is considered a synonym of *A. phalloides* var. *alba* [76,77], the name *A. decipiens* would be available for the KOH+ specimens (see Discussion below). However, we think that this option would not contribute to stabilizing the nomenclature of this group, and the interpretation of *A. verna* as a KOH+ species is here preferred. Two samples of *A. verna* collected in Oise department, 50–60 km north of Paris, were found in Romagnesi’s herbarium: HR56.31 from Apremont and HR71.42 from Coye-la-Forêt. To close the debate and fix the nomenclature of *A. verna*, we epitypify this name with collection HR56.31.

The variability in spores of *A. verna* caught the attention of various mycologists, from Gilbert [78] to Malençon & Bertault ([79]; Figure 7) or Neville & Poumarat [38]. In the present study, no spore print was available, and spores were observed on exsiccata as deposits at the apex of the stipe or near the edge. Most collections (including the epitype) studied here displayed only a moderate variability of spore shape, which ranged from subglobose to ellipsoid (Figure 6A,C). Specimens with narrowly ellipsoid to cylindrical spores such as those described as *A. verna* f. *ellipticospora* by Gilbert [78] or those described by Bertault [80] from specimens collected in Morocco, could not be revised here and a special attention should be given to such collections in the future. The description of *A. gilbertii* f. *subverna* Bertault & Parrot from Morocco, about which Bertault [81] wrote: “seules les spores cylindriques permettent de les séparer” (only the cylindrical spores make a difference [with *A. verna*]), suggests that this taxon, which Neville & Poumarat [38] validated by selecting a holotype found in the island of Porquerolles (south eastern France), could be related also to *A. verna*.

Although *Amanita verna* is univocally described in the literature as having a white to more or less ochre pileus, several collections from Italy with a greenish yellow pileus have been found by Raumi [82], described under the provisional name ‘*Amanita verna* f. *xanthoviridis*’ ad. int. The ITS rDNA, TEF1, and RPB2 sequences from one of these collections (ALV21204, kindly shared by M. Raumi, 1 pileus deposited at LIP n°0002275) do not differ from those of typical samples of *A. verna* (Figure 1). The yellow tinge seems to be due to the presence in the subpellis of radially oriented hyphae 5–6.5 µm wide with thick, yellowish walls (up to 1 µm thick), which were not observed in white specimens; the spores are in the range of the type, [1/22] 9.2–10.4 × 7.0–8.2 µm, Q = 1.2–1.3, Q_av_ = 1.2, mostly broadly elliptical, with a few macrospores likely issued from 2-spored basidia, up to 13.5 × 10.5 µm. This remarkable form might be confused with *A. phalloides*, from which it differs by the fruiting season and the strong yellow reaction of all parts of the basidiome to KOH (Figure 2). No complete specimen could be obtained for the purpose of this article, and thus a full description of this form and the validation of its provisional name could not be provided here.

5.***Amanita vidua*** Gasch, G. Moreno & P.-A. Moreau, sp. nov., Figure 2F,H,J, Figure 4A,B and Figure 7

MycoBank number: MB 842793

Misapplied names: *Amanita verna* sensu Bertault [80,83], *Amanita verna* var. *verna* sensu Trimbach [84], La Chiusa [85], Neville & Poumarat [38], etc.; *Amanita tarda* sensu Contu [86]

Etymology: from Latin noun *vidua*: widow, an ironic reference to its high toxicity and to the fact that this species became ‘widowed’ after the name *A. verna* could not be employed for it anymore.

Diagnosis: Differs from *Amanita verna* by the absence of yellow reaction with KOH on all tissues, as well as its less distinctly bulbous stipe, and often umbonate pileus early tinged with ochraceous tones. Typically occurring in spring in siliceous soils of the Mediterranean regions, under *Quercus ilex* or *Q. suber*.

Holotype: SPAIN: Andalucia, Galaroza, under *Quercus ilex* on siliceous soil, 4 May 2019, A. Gasch & P.-A. Moreau, PAM19050401 (coll. P.-A. Moreau, LIP 0401591).

Description: Basidiomata solitary to scattered. Pileus at first subglobose, then expanding to flat-convex or broadly convex-umbonate, pileipellis easily detachable, smooth, shiny, thin, off-white to cream-white, sometimes developing ochre-yellow or ochre-pinkish tones at the centre, or uniformly isabelle ochre-brown (reminding of A. eliae) when old or dry, lacking adnate fibrils; margin not striated, inrolled at first, soon expanded, persistently white. Lamellae pure white, crowded, free with broad collarium at maturity, with lamellulae, white; edge floccose to serrulate, white, becoming cream-yellow to salmon ochre towards margin when old. Spore print white. Stipe 7.5–16 × 1–1.8 cm, cylindrical to subcylindrical, white, smooth, pruinose-floccose at the apex and slightly striated above the ring, somewhat zebrate below, with a slightly bulbose base about 2.5 cm diam., with a sack-like membranous white volva at the base, usually buried, without internal limb; partial veil forming a descending ring, membranous, ample but early adpressed on stipe and collapsed in adults, thick, white, often doubled by an oblique bandlet some mm below. Universal veil without remnants on the pileus. Flesh white, becoming cream to pinkish with age. Odour fungoid, not remarkable, even with alteration. Taste fungoid. Reaction to 10% KOH pale ochre on pileipellis, nil elsewhere.

Spores [6/140] variable in shape, from subglobose (20–60%) to broadly ellipsoid (30–50%) and ellipsoid (10–40%), measuring (8.8–) 9.0–10.8 (–11.2) × (6.8–) 7.2–8.5 (–9.1) µm, Q = 1.15–1.30 (–1.60), Q_av_ = 1.2 [from the holotype], smooth, hyaline content before maturity, some spores filled with yellowish droplets with age, amyloid. Basidia 4-spored, cylindrical with tapering base, measuring 32–58 × 9–12 µm, clampless. Subhymenium 30–50 µm thick, nearly pseudoparenchymatous with lobate, constricted or cordiform elements measuring 15–25 × 12–20 µm. Hymenophoral trama with a mediostratum 120–150 µm thick in fresh specimens, collapsing to 40–50 µm in herbarium samples, filamentous, made up of parallel slender hyphae 2.5–5 µm wide mixed with broad, inflated hyphae 7–11 µm wide; hymenopodium not differentiated. Lamellae edges fertile (mature specimens); terminal cells sparse, cylindrical to clavate-capitate, measuring 30–38 × 11–14 µm. Pileipellis thick, 2-layered; suprapellis a 30–40 µm thick ixocutis formed by gelatinized filamentous hyphae 2–5 µm diam., clampless, some of them with yellow granular content in KOH; subpellis not gelatinized, otherwise similar to suprapellis, 30–50 µm thick, mostly made up of parallel, slender hyphae 1–2.5 µm wide and also sparse broader hyphae up to 5 (–7) µm wide, all of them smooth with slightly thickened, yellowish walls. Partial veil (ring) made of intricate cylindrical hyphae 2–8 µm wide, smooth, clampless, with walls thickened up to 1.5 µm; upper surface filamentous with sparse sphaeropedunculate elements measuring 12–15 × 9–11 µm. Volva distinctly 2-layered; inner layer filamentous, about 150–300 µm thick, made up of intricate cylindrical hyphae 3–7.5 µm diam., often branched, clampless, mixed with very rare sphaerocytes or inflated elements up to 45 × 22 µm, all smooth and colourless; outer layer up to 40 µm thick, made up of cylindrical slender hyphae 3–7 µm wide, smooth with slightly thickened walls (up to 0.2 µm), yellowish in KOH, locally incrusted with yellowish deposits.

Habitat and distribution: infrequent vernal species that probably establishes ectotrophic mycorrhizae with Mediterranean species of *Quercus* (*Q. ilex* ssp. *ballota*, *Q. pyrenaica*, *Q. suber*, etc.), rarely also with other trees, i.e., *Castanea*, *Fagus*, *Pinus*. Known from acidic soils of the Mediterranean basin, from Spain to Lebanon, as well as Morocco [76], from April to early June.

Examined material: FRANCE: Var, la Garde-Freinet, under *Quercus suber* and *Castanea sativa* on siliceous soil, leg. S. Kizlik & Société Mycologique de Provence, 7 June 1987, La Garde Freinet nº83, 0002253 (LIP). Var, Pourrières, *Quercus ilex* in garrigue, M. Bon, November 1976, herb. M. Bon 761,022 (LIP, not sequenced). ITALY: Lazio, Anguillara Sabazia, among litter in acidic soil, along a sidetrack on a steep slope facing north in a pure stand of *Quercus ilex*, 185 m a.s.l., ca 100 m from the shore of lake Bracciano, 16 May 2021, leg. M. Gelardi, F. Costanzo & O. Gelardi, MG846, 0402256 (LIP). LEBANON: Bnebil, Mount-Lebanon, in a mixed stand of *Pinus pinea* and *Quercus infectoria*, 1000 m a.s.l., 18 April 2018. SPAIN: Madrid, Chapineria, under *Quercus ilex* subsp. *ballota*, leg. Soc. Micol. Madrid, 14 Apr 2002, 31,984 (AH). Madrid, leg. Soc. Micol. Madrid, 5 May 2002, 31,988 (AH). Madrid, El Pardo, under *Quercus ilex* subsp. *ballota*, leg. V. Sánchez, 24 May 2008, 31,838 (AH). Idem, 2 June 2008, 31,841 (AH). Segovia, Revenga, under *Quercus ilex* subsp. *ballota*, leg. G. Moreno, F. Esteve-Raventós, A. Sánchez, J. Díez, V. D’Angelo & M.M. Dios, 5 June 1998, 31,989 (AH). Madrid, Sierra de Guadarrama, under *Quercus pyrenaica*, leg. Soc. Micol. Madrid, 22 April 2001, 31,990 (AH). Madrid, Villa del Prado, under *Quercus ilex* subsp. *ballota*, leg. G. Moreno, 20 April 2002, 19,696 (AH). Idem, leg. J.C. Campos, 25 April 2003, 37,756 (AH). Idem, 37,757 (AH). Idem 37,758 (AH). Idem 37,759 (AH). Idem, 20 April 2004, 37,760 (AH). Idem, 37,761 (AH). Idem, leg. J. Hernanz, J.C. Campos & J. Campos, 1 May 2004, 37,727 (AH). Idem, 37,728 (AH). Idem, leg. J.C. Campos, 18 May 2008, 19,697 (AH). Idem, 31,837 (AH).

Notes: *Amanita vidua* is very similar to *A. verna*, both sometimes fruiting in the same areas. No evident diagnostic morphological features could be found to distinguish these species, which differ mainly in their reaction to KOH (chrome yellow in *A. verna*, negative to faintly ochre in *A. vidua*), in the narrower and less differentiated bulb at the base of the stipe, and in the exclusively Mediterranean distribution of *A. vidua*, which does not reach temperate areas where only *A. verna* is present. Genetically, ITS rDNA, rpb2, and tef1 sequences (introns excluded) of *A. vidua* are 94–96% similar to those of *A. verna* or *A. phalloides*, while 28S rDNA is 99–100% similar to those of these species.

6.***Amanita virosa*** Bertill., Dict. Encyclop. Sci. Médic. (Paris) 1(3): 497 (1866) [nom. nov. based on *Ag. virosus* Fr. 1838, illegit.], Figure 2D, Figure 3B, Figure 4D and Figure 8

Basionym: *Agaricus virosus* Fr., Epicr. Syst. mycol. (Upsaliae): 3 (1838), illegit. [non-*Ag. virosus* Sow., 1809]; *Amanitina virosa* (Bertill.) E.-J. Gilbert, in Bresadola, Iconogr. mycol., Suppl. I (Milan) 27: 78 (1940)

Typification: no type designated by Fries [8]; no original material known to us. Neotype here designated: SWEDEN: Fiby Urskog, old-growth *Picea abies* forest, leg. A. Dahlberg & P.-A. Moreau, 12 August 2006, coll. P.-A. Moreau SU12-06-19, 0402243 (LIP), MycoBank MBT 10005510.
=*Amanita virosa* var. *aculeata* Voglino, Boll. Soc. bot. ital., 1894: 120 (1894)

Description of the neotype: Pileus 4–11 cm, cylindrical to conical-obtuse at first, later flexuose with the center persistently convex or sometimes umbonate, never flattened or depressed, greasy to almost viscid, shiny-silky when dry, pure white, disc slightly ochraceous when old; margin incurved for a long time, then irregularly expanded, smooth, never striated, white, usually with traces of a floccose partial veil hanging. Lamellae crowded, rather thick, free, collariate at maturity, pure white, with subtle pinkish tones when mature; edge thick, heavily floccose, usually with adherent floccose patches of the partial veil. Stipe 7–18 × 0.6–1.8 cm, cylindrical or narrowed towards the apex, equal, slender, often curved at maturity, inflated at the base with a sphaerical bulb up to 3 (–4) cm wide; surface entirely floccose, with bandlets or ascendant scales on the lower part with age, somewhat striate at the apex, pure white and remaining so with age; partial veil membranose-floccose, fragile, easily dissociated, usually partly hanging high on stipe, often absent in mature specimens, partly adherent to lamellae and to the pileus margin, white; universal veil forming a resistant membranous volva, white to somewhat ochraceous on surface. Context white, unchanging with age. Herbarium samples remaining white, or faintly pale yellow at the center of the pileus; fresh specimens lack any smell, but they can develop a fungoid, unpleasant odor (especially in the bulb) with age; taste mild, fungoid. Reaction to 5% KOH chrome yellow on all surfaces and context except the inner layer of the volva.

Spores [2/78] mostly subglobose (50%), also broadly ellipsoid (30%) and rarely globose or ellipsoid (<10% each), (7.0–) 8.1–10.2 (–11.7) × (5.6–) 7.0–8.9 (–11) µm, Q = 1.1–1.2 (–1.4), Q_av_ = 1.1, smooth, amyloid, with a dense yellow content in KOH at maturity, thin-walled and mostly collapsed on exsiccata. Basidia 4-spored, 32–52 × 9–12 µm, clavate, with a dense yellowish content. Subhymenium poorly developed, 10–15 µm wide, made up of small polygonal elements. Hymenophoral trama with a mediostratum made up of slender hyphae 2–7 µm wide mixed with numerous physaloid hyphae 18–25 µm wide arranged in diverging orientations; hymenopodium not differentiated. Lamellae edges thick, floccose, similar in structure to the partial veil. Pileipellis an ixocutis 100–120 µm thick, made up of slender hyphae measuring 2–4 µm wide, towards subpellis mixed with broader hyphae with yellow content, up to 9 µm wide. Subpellis not gelatinized, 60–80 µm thick, with parallel hyphae mixed with numerous oleipherous hyphae 3–4 µm wide with homogeneous yellow content in KOH. Context of the pileus distinct from subpellis, made up of intricate hyphae, often inflated up to 15 µm wide, without oleiferous hyphae. Partial veil homogeneous, pulverulent, made of mostly short cylindrical elements measuring 15–25 × 9–11 µm, mixed with cylindrical hyphae 4–7 µm wide, with slightly thickened yellowish walls. Universal veil 2-layered: outer layer thin, irregular, made up of hyphae embedded in a deep yellow extraparietal amorphous pigment, strongly thick-walled, wall up to 2 µm thick, deep yellow, with frequent cylindrical terminal elements, sometimes lobate, with rounded apex up to 6 µm wide; inner layer made up of rather broad hyphae, 7–14 µm wide, with very rare large sphaerocysts up to 85 × 55 µm, intricate, becoming more parallel towards the outer layer; no oleipherous hyphae observed.

Notes: In Europe, *Amanita virosa* cannot be easily confused with other species, being the only *Amanita* sect. *Phalloideae* growing on acidic organic soils, with a creamy ring and conical pileus. However, although being a strictly Eurasian species (as suggested by the sequences available in GenBank and UNITE databases), its name has been misapplied to various white extra-European species. Its neotypification with a collection from Sweden will fix the name and avoid further misidentifications.

Examined material: FRANCE: Nord, Condé-sur-Escaut, forêt domaniale de Bonsecours, 15 September 2013, R. Courtecuisse & P.-A. Moreau, PAM13091506 (LIP), not sequenced.

Other species examined: ***Amanita citrina*** Pers. Tent. disp. meth. fung. (Lipsiae): 66 (1797) (as *Amanita phalloides* var. *larroquei* F. Massart & Beauvais ex F. Massart, Bull. Soc. linn. Bordeaux 31(4): 223 (2004)). FRANCE: Landes, Bombannes, under *Pinus pinaster* in sand dune, Massart 05008, 0402241 (LIP, authentic material of *A. phalloides* var. *larroquei*). ***Amanita strobiliformis*** (Paulet) Bertill. Dict. Encyclop. Sci. Médic. (Paris) 1(3): 499 (1866) (as *Amanita verna* ss. Romagnesi [71]). FRANCE: Manche, Saint-Sauveur-le-Vicomte, église de Taillepied, in sand and gravels along a broadleaved wood, de Bernon & Guérin, 17 Aug 1983, coll. H. Romagnesi HR83.65a and HR83.65b (PC, as ‘*A. verna*’).

 Key to European species of sect. *Phalloideae*1.KOH reaction bright yellow on surfaces…………………………………………………………21.KOH reaction none or only pale cream yellow…………………………………………………52.Stipe smooth. Ring complete, not lacerated at maturity, membranous, persistent on stipe, made up of mostly filamentous hyphae…………………………………………………………32.Stipe woolly-scaly. Ring thin, usually lacerate, hanging at margin or disappearing at maturity, mostly made up of short cylindrical elements. Smell none till late maturity, finally slightly nitrous. Spores mostly subglobose (Q_av_ < 1.15). Moist, acidic, often peaty forests under conifers (*Pinus, Picea*) or broadleaved (*Betula, Fagus*)…………………*A. virosa*3.Pileus soon flattened to depressed. Smell not perceptible in the bulb. Basidiomata white to ochraceous when dry [pileus yellowish green in ‘f. *xanthoviridis’*]. Subhymenium 30–45 µm thick, of puzzle-like structure. Spores mostly broadly elliptical (Q_av_ > 1.15). Calcareous or mineral-rich soils, mostly under *Quercus* spp., usually in spring……………………*A. verna*3.Pileus conical to campanulate……………………………………………………………………44.Strong smell of iodine in the bulb. Basidiomata turning uniformly lemon yellow when dry. Subhymenium poorly developed, ramose, less than 20 µm thick. Spores mostly subglobose (Q_av_ < 1.15). Acidic, mesophilic forests, mostly under *Quercus*…………*A. amerivirosa*4.Smell not of iodine. Basidiomata turning brownish when dry. Subhymenium well-developed, with puzzle-like elements. Spores mostly broadly ellipsoid (Q_av_ > 1.15). Mediterranean forests…………………………………………………*A. phalloides f. porrinensis*5.Fruiting in spring or early summer. KOH reaction dirty cream-yellow. Pileus white turning ochraceous at disc when old or dry, often gibbose at maturity. Under oaks on mineral-rich soils, in spring, Mediterranean…………………………………………*A. vidua*5.Fruiting in autumn. KOH reaction nil or pale yellowish. Pileus usually greenish with dark fibrils (typical form), but also yellowish, ochraceous, greenish brown to pure white or with only faint greenish shade, convex to flattened but only exceptionally gibbose (*A. phalloides* f. *porrinensis*). Under various deciduous (especially *Fagaceae*) or coniferous (*Abies, Cedrus, Pinus, Picea*) trees on acidic to moderately calcareous soils, summer to early winter, Mediterranean to cold-temperate…………………………………………*A. phalloides*

### 3.3. Toxicology

Results from the HPLC analysis (Figure 9) show that the contents of amatoxins and phallotoxins are strongly associated with the genetic identity of the samples. *Amanita virosa* contains α-amanitin, phallacidin, and phalloidin but lacks β-amanitin. This is partially in accordance with the literature [62,87,88,89]. However, according to other authors, some samples of *A. virosa* from North America [90,91] or Japan [92] contain β-amanitin, but no sequencing was performed to check the actual identity of these samples, which could represent other taxa. Samples of *A. amerivirosa* analyzed in the present work do not show amanitins or phallacidin, resembling the profile exhibited by other samples of *A. virosa* from North America [90]. Samples of *A. verna* analyzed in the present work contain large amounts of α- and β-amanitins (5.47 and 10.26 mg/g dry matter, respectively) and phallacidin (4.69 mg/g dry matter), and a lower amount of phalloidin (0.64 mg/g dry matter) in accordance with other works [25,89]. With regard to *A. phalloides*, different contents of α-amanitin were found in samples LIP:0402239 (from a *Pinus pinaster* forest with sandy soil) and SMM2018-10 (from a *Quercus ilex* stand with *Arbutus unedo*). Soil properties are thought to influence the toxicological profile of *A. phalloides* [58,93], and this could explain the differences found. However, Bon & Andary [94] did not find α- or β-amanitins by thin layer chromatography (TLC) in the type specimen of *Amanita dunensis* (=*A. phalloides*). The method of extraction and/or quantification (HPLC vs. TLC) might account for these differences. In addition, what remains of *A. dunensis* type collection at LIP (Bon 4119) is a half-specimen in bad condition, probably already very mature when collected. This could be another factor that influences the toxicological results. More samples from the Atlantic coast should be analyzed to check whether actual differences exist or not. Finally, the analyses conducted on specimens of *A. vidua* PAM1905041 at various developmental stages revealed low inter-specimen variations. Large amounts of all amatoxins (α-amanitin: 4.73–7.18 mg/g dry matter, β-amanitins: 5.02–8.50 mg/g dry matter) and phallotoxins were detected (phallacidin: 6.53–7.78 mg/g dry matter), suggesting that *A. vidua* is highly toxic and must be considered as another deadly ‘destroying angel’, maybe the most poisonous of the entire group.

### 3.4. Epidemiology

The analysis of records obtained from Spanish databases was divided into two periods: (1) spring period (from 1 February to 30 July), and (2) fall period (from 1 August to 31 January). These periods are clearly separated by the cold or dry periods when *Amanita* species do not fruit, i.e., mid-winter (January), in which no case of phalloidian poisoning was registered in any database, and mid-summer (July) with only one case known, attributable to *A. verna*. Ninety-four fungal poisoning clusters (≥2 patients), representing 406 symptomatic cases, were retrieved from the RENAVE database (1980–2020, source 1). Five clusters in the spring period (Feb–Jul) and ten clusters in the fall period contained cases compatible with phalloidian poisonings. Spring clusters led to 10 hospitalizations without registered deaths and originated in the provinces of Aragon (2 clusters, 5 cases), Andalusia (2 clusters, 3 cases), and Castile and León (1 cluster, 2 cases). During the fall (August-January), 10 clusters were retained as possibly phalloidian, representing 28 hospitalizations and 2 deaths. One fall case was attributed explicitly to *Lepiota brunneoincarnata*, while the others were either attributed to *Amanita* spp. or not identified in RENAVE; fall intoxications occurred in Andalusia (3 clusters, 9 cases), Castile-La Mancha (2 clusters, 4 cases), Catalonia (2 clusters, 4 cases), Aragon (1 cluster, 2 cases), Galicia (1 cluster, 2 cases), and Castile and León (1 cluster, 2 cases).

The CMBD (1997–2015) and RAE-CMBD (2016–2020) databases (source 2) contain 1817 cases of putative fungal intoxications, 219 of them presenting acute hepatitis, characteristic of the phalloidian syndrome. During the spring period (Feb–Jul), 16 cases were recorded, which originated in Andalusia (4), Extremadura (3), Madrid (2), Castile and León (2), Castile-La Mancha (2), Catalonia (2), and Aragon (1). No spring cases were recorded in 10/17 regions, including all those in the northwest, the Spanish Levant, and the Islands. During the fall (Aug–Jan), 203 cases were recorded all over Spain, mainly occurring in the Catalonia (64), Castile and León (37), Madrid (27), Galicia (17), Aragon (12), and Andalusia (9) regions; Murcia is the only community without any phalloidian cases in the fall. It is noticeable that all spring cases in Andalusia occurred in the western part of the region (Huelva, Seville) close to Extremadura, which has a spring incidence nearly 10 times higher than the national average. Global cumulative incidence of phalloidian poisonings in Spain for the 1997–2020 period was 4.8/1,000,000 hab. (0.34 in spring and 4.34 in fall), calculated from the population in the middle of the period (detailed for each community in Figure 10), and average cumulative incidence was 0.20/1,000,000 inhabit. per year.

## 4. Discussion

*Amanita verna* was originally described by Bulliard ([5], pl. 108) under the trinomial name *Agaricus bulbosus vernus*. The original plate illustrates four complete, entirely white basidiomes and a longitudinal section of the biggest one. In the extensive legend of the plate, the author describes a spring-fruiting *Amanita* with pure white, convex to depressed pileus, rounded bulb, and a well-formed membranous ring placed high on a smooth stipe. Bulliard indicates that the species is common in the woods around Paris, and cites a greenish form, which he later described in detail (pl. 506) under the name *Agaricus bulbosus* Bull. Shortly after this publication, the taxon was upgraded to species status by Lamarck [95] as *Amanita verna* (Bull.) Lam., with a similar description.

The history of the name *Amanita verna*, sanctioned by Fries [96] at species rank, has been reviewed by various authors, with opposing conclusions about the real identity of Bulliard’s species, according to personal interpretations of the aforementioned elements (Table 1). Konrad & Maublanc [76], for instance, interpreted *A. verna* as the pigmentless form of *A. phalloides*, also named *A. phalloides* var. *alba* or *A. andaryi* (invalid name), based on the fact that Bulliard included in his *Ag. bulbosus* white and yellow specimens without other distinction. However, most authors accepted *A. verna* as a vernal species, much rarer than mentioned by Bulliard (at least in Northern France), which differs from *A. virosa* by its smooth stipe.

In a posthumous publication (a catalogue of macrochemical reactions of many species without descriptions), Bataille [77] revealed a new feature observed in specimens identified by him as *A. verna*: the absence of reaction to 10% KOH on the pileus, in contrast to *A. virosa*, which turned bright chrome-yellow to this reagent. However, many authors after him reported a positive reaction on their own collections of *A. verna* [67,86,97], among others, and so did Trimbach [84], who accommodated the KOH+ samples in a separate variety, *A. verna* var. *decipiens* Trimb., expressing his own doubts about the taxonomic value of the KOH reaction. A couple of years later, Trimbach coined the name *A. verna* var. *tarda* for a variety found in Switzerland with less truncate lamellae, fruiting in autumn, and lacking any visible reaction to KOH.

Bertault [80,83] reported KOH- collections of *A. verna* with more or less uncinate lamellae in Mediterranean cork oak (*Quercus suber*) forests of Morocco. Contu [84] applied the name *Amanita tarda* to this taxon, retaining the name *A. verna* for the KOH+ species. The truncate lamellulae emphasized by Trimbach [98] for *A. verna* var. *decipiens* were discussed and re-evaluated by Russi & Josserand [97] in a detailed study of KOH+ *A. verna* from the Lyon area. Specimens of KOH- *Amanita verna* were known only from southern Europe and Morocco, until Romagnesi [75] published an initially convincing record from Normandy, northern France (coll. HR83.65). However, after a long discussion, Romagnesi concluded that KOH- *A. verna* was a thermophilic species unlikely to have been seen by Bulliard around Paris, suggesting keeping the name *A. verna* for the KOH+ species, and applying the name *A. verna* f. *ellipsospora* E.-J. Gilbert to his KOH- collection. Neville & Poumarat [38] again considered *A. verna* as KOH- after checking three collections sent to them from southern France and employed the name *A. verna* var. *decipiens* for KOH+ collections.

In the present work, KOH+ and KOH- specimens of *A. verna* are shown to belong to distinct species. KOH- specimens are typically found in Mediterranean habitats of southern Europe (apparently only in acidic soils), while KOH+ samples are found in Mediterranean and temperate climates. Unfortunately, we could not find any preserved specimens of *A. verna* from near Paris after 1971. Guy Redeuilh (pers. comm. to P.-A. Moreau) reported that *A. decipiens*/KOH+ *A. verna* was usually collected in some relict oak woodland in sandy places along the Seine River some 30 years ago; the extensive urbanization during the last century destroyed most of the favorable places where Bulliard might have observed it. One KOH+ collection from Oise (North of Paris), HR56.31, is proposed here as an epitype to fix this ancient taxonomic issue. *Amanita verna* var. *tarda* is interpreted as a synonym of *A. phalloides* var. *alba*, as suggested by Neville & Poumarat ([38], pag. 567); Romagnesi also misidentified one of his collections of *A. phalloides* var. *alba* as *Amanita verna* (PC:HR60.109).

The KOH- samples are here named *A. vidua*, since they lack any other existing name. The spectacular yellow reaction to KOH displayed by *A. verna* is present also in *A. virosa* and *A. amerivirosa*, and therefore this macrochemical feature could have been present in the common ancestor of the whole ‘North Hemisphere lineage’ but lost in *A. phalloides* (although somewhat present in f. *porrinensis*) and strongly weakened in *A. vidua*. This reaction appears under optical microscope to be mainly located at the abundant epiparietal mucoid deposits on most hyphae, and extends to spore content, which shows a yellowish lipidic content in 5% KOH, in all species with the exception of *A. phalloides*, and is only scarcely visible in *A. vidua*. The spores themselves have a distinctive shape, mainly subglobose in *A. amerivirosa* and *A. virosa* (Q_av_ = 1–1.15), mainly broadly ellipsoid with a higher Q_av_ (1.13–1.22) and usually narrower (<7.8 µm wide) in *A. phalloides* and *A. vidua*. *Amanita verna* has a remarkable variability in spore shape, observed among different collections as described above (see Figure 6A,C,D), but also among spores obtained from the same basidiome as illustrated by Malençon & Bertault [79].

The four toxins studied in the present work are apparently occurring across the whole sect. *Phalloideae*, suggesting that all species are probably toxic, but there are some remarkable differences among them. In the virosa clade, β-amanitin is not detectable or at a very low concentration, and *A. amerivirosa* has a very low content of α-amanitin, completely lacking phallacidin. In addition, chromatograms of *A. amerivirosa* display some peaks absent in *A. virosa*, with UV spectra corresponding to compounds lacking 6-hydroxytryptophan [62]. The identification of these compounds would be of great toxicological interest. The actual amount of toxins is highly dependent on the extraction conditions, with major impacts of solvent, temperature, and processing times during the maceration step. Freshness of the samples analyzed also influences the results of HLPC, but little information exists about the degradation of amatoxins and phallotoxins during storage. Stijve & Seeger [99] observed a decrease in amanitins (α- and β-) and phalloidin contents over time in specimens collected in the same location in different years and suggest that phallotoxins might be less stable during storage than amanitins, whose stability is known to depend on treatment and storage of the samples [100,101,102]. Results obtained in the present work support this hypothesis, since at least α- and β-amanitins seem less abundant in older specimens (i.e., *A. vidua* from 1987 vs. *A. verna* from 2001). However, these observations were not the purpose of the present work, and therefore a more detailed study should be carried out to further clarify this issue.

*Amanita amerivirosa* has been reported from western France for more than 30 years, first as *A. decipiens* [103], then as ‘American *A. virosa*’, and finally named *A. virosa* var. *levipes* [38]. Carefully surveyed by local mycologists who could follow its fast expansion from the Nantes area to Bordeaux and Normandy [104,105], it has become especially abundant in the last 15 years, recently reaching the Paris area [106]. Fortunately, in spite of the abundance of this species in western France, the number of poisoning reports caused by *A. amerivirosa* in France is still limited [2,107], with only two cases confirmed in 2018 (identified as *A. virosa* var. *levipes*; Moreau, unpublished data), and they lacked associated symptomatology. This could be related to the low amount of toxins detected in the collections of *A. amerivirosa* studied by Beutler & Der Marderosian [90], Bonnet & Basson [91], and the present work. However, this needs to be confirmed by additional analyses, and so this species should not be excluded from the list of lethal ‘destroying angels’ for the time being. On the other hand, *A. vidua* and *A. verna* are evidently highly poisonous on the basis of the amatoxins detected in these species. Both species may grow in the same sites and season as *A. ponderosa* Malençon & R. Heim (the so-called “gurumelo”), a highly prized edible fungus endemic in southern Portugal and south-western Spain (Huelva, Badajoz, Cáceres; [74,108]). This could explain the abnormally high incidence of spring intoxications compared to fall intoxications in the Spanish provinces of Extremadura and Andalusia (Figure 10). *Amanita boudieri* Barla (*Amanita* sect. *Lepidella*), which can cause another lethal syndrome (acute renal failure; [109]), has also been mistaken for *A. ponderosa* but differs morphologically by the absence of a membranous volva and ring [108].

The average incidence of phalloidian poisonings (mostly due to lethal *Amanita* species) in Spain during the period analyzed in the present work was 0.20/1,000,000 inhabit. per year. This value is comparable to botulism (0.20 per year in the last decade; [110]), another food-borne disease, and thus represents a comparable challenge for public health. More detailed and homogeneous data on similar poisoning cases in Europe, as well as more precise identification of the responsible species, would be necessary to evaluate and manage the risk, which seems to differ between regions. Especially, information campaigns targeting unexperienced mushroom collectors should warn them of the presence of the lethal ‘destroying angels’. Since *A. vidua* can be found in the whole Mediterranean basin (from Spain to Levant, and probably also in North Africa) and shows a high toxicity, a special focus should be placed on this species in the entire area. However, spring intoxications are apparently not relevant in some areas, such as Apulia (southern Italy; [111]), suggesting that either *A. vidua* and *A. verna* are not present there, or else that in the absence of similar edible early-fruiting species or of local picking traditions attached to them, they are not accidentally consumed.

## 5. Conclusions

(a)Five distinct species of *Amanita* section *Phalloideae* can be found in Mediterranean Europe: *A. phalloides*, *A. virosa*, *A. verna*, *A. amerivirosa*, and *A. vidua* sp. nov., a new name proposed for the KOH-negative Mediterranean species previously described as *A. verna* or *A. decipiens* by various authors.(b)Three taxa, *Amanita decipiens*, *A. porrinensis,* and *A. virosa* var. *levipes* are here considered later heterotypic synonyms of *A. verna*, *A. phalloides,* and *A. amerivirosa*, respectively.(c)Samples identified with the provisional name ‘*Amanita verna* f. *xanthoviridis*’ are not genetically different from *A. verna* f. *verna*, suggesting that this species can produce distinctly colored basidiomata.(d)All Mediterranean species of *Amanita* sect. *Phalloideae* contain amatoxins and phallotoxins, although their identity and quantity differ between species. The newly described *A. vidua* is the most amanitin-rich European species of the section. No amatoxin was detected in the analyzed samples of *Amanita amerivirosa* from western France.(e)In Spain, the incidence of mushroom poisonings in spring attributed to species of *Amanita* sect. *Phalloideae* is about 10 times higher in regions where other similar edible species are collected in the same season, representing up to 1/3 of all mushroom poisoning during the year in these regions.

## Figures and Tables

**Figure 1 biology-11-00770-f001:**
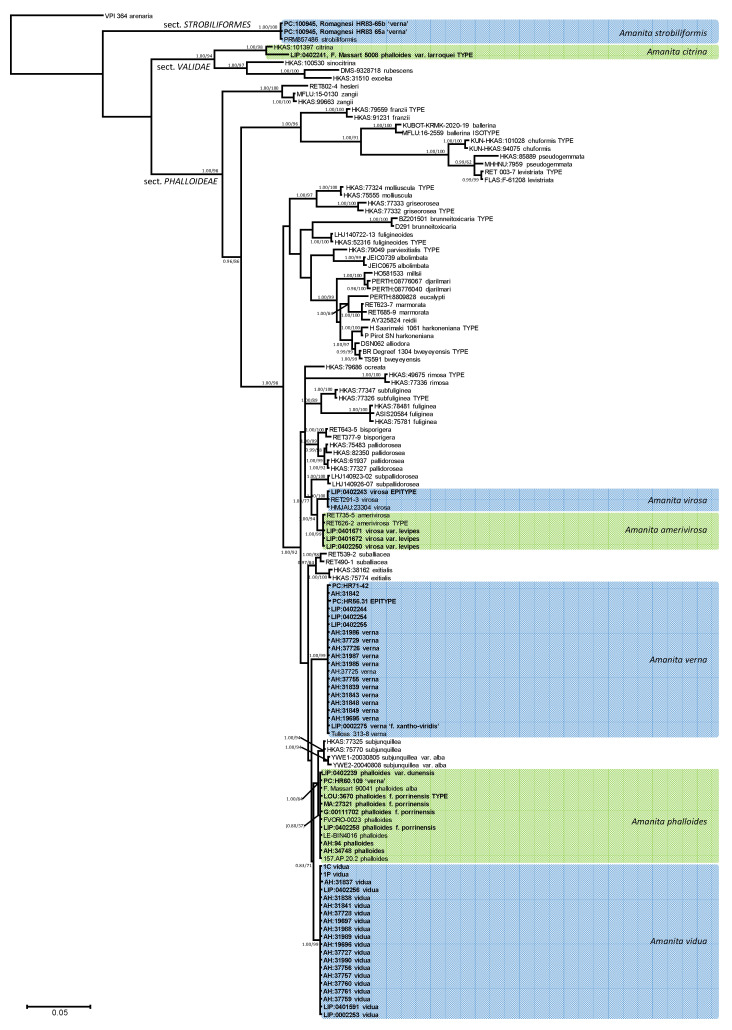
A 50% majority rule ITS rDNA-28S rDNA consensus phylogram of *Amanita* sect. *Phalloideae* and selected lineages of the neighboring sections *Strobiliformes* and *Validae* (with *Amanita arenaria* from sect. *Arenaria* [66] as outgroup) obtained using MrBayes from 7050 sampled trees. Nodes were annotated if they were supported by ≥0.95 Bayesian posterior probability (left) or ≥70% maximum likelihood bootstrap proportions (right). Non-significant support values are exceptionally represented in parentheses. Sequences newly generated in this study are in bold face.

**Figure 2 biology-11-00770-f002:**
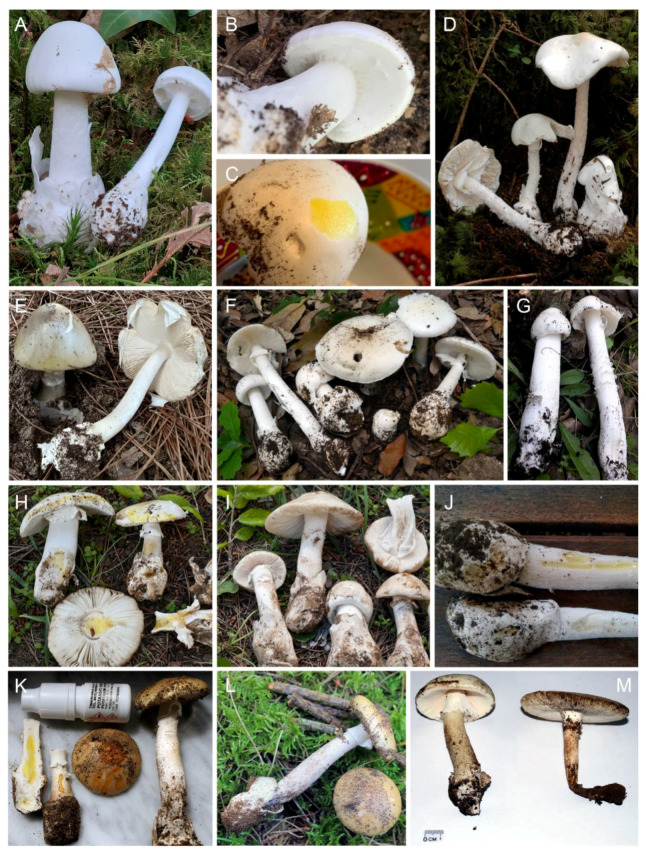
Macroscopical features of ‘destroying angels’ (**A**,**B**). *Amanita amerivirosa*. (LIP:0402252), photo P.-A. Moreau. (**C**) *Amanita amerivirosa*, reaction to 10% KOH (LIP:0402250), photo N. Journe. (**D**) *Amanita virosa*, neotype (LIP:0402243), photo P.-A. Moreau. (**E**) *Amanita phalloides* (LIP:0401670). (**F**) Mixed basidiomes of *Amanita verna* (LIP:0402254) and *A. vidua* (LIP:0402256), photo M. Gelardi. (**G**) *Amanita phalloides* f. *porrinensis* (LIP:0402258), photo A. Mua. (**H**) *Amanita verna* (AH:31849), photo G. Moreno. (**I**) *Amanita vidua* (AH:19696), photo G. Moreno. (**J**) *Amanita verna* (LIP:0402254, top) and *A. vidua* (LIP:0402256, bottom) reaction to KOH, photo M. Gelardi. (**K**,**L**) *A. verna* ‘f. *xanthoviridis*’ ad. int. (LIP:0002275), photo M. Raumi. (**M**) *Amanita vidua* from Lebanon (1C and 1P).

**Figure 3 biology-11-00770-f003:**
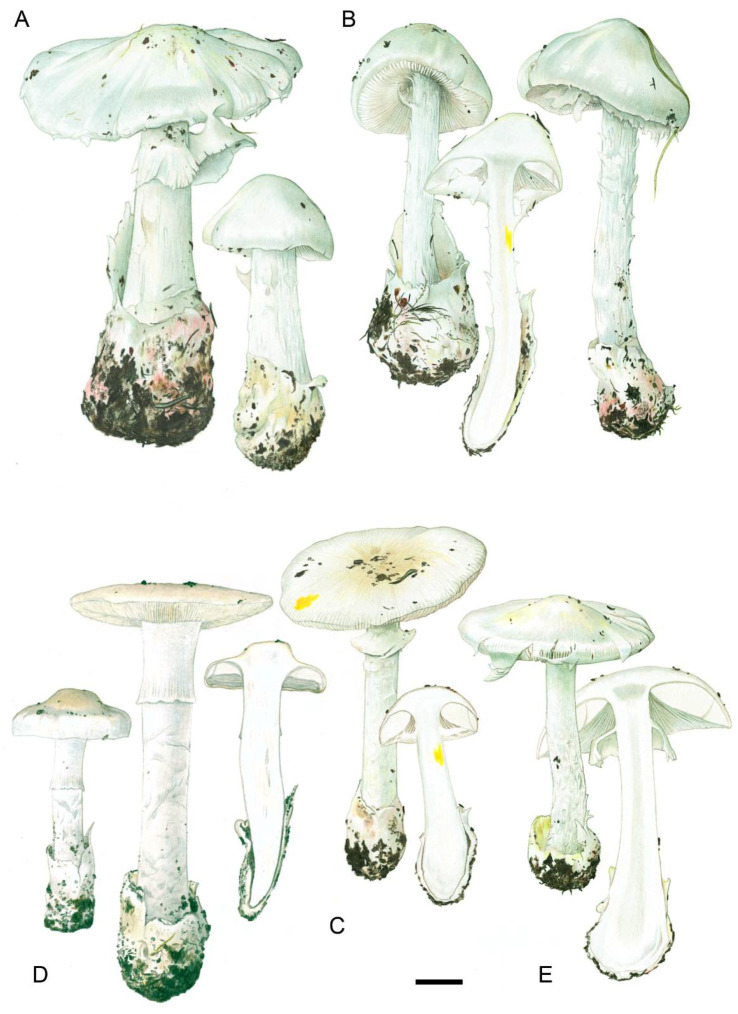
Drawings of basidiomes in Reumaux & Carteret (2013). (**A**) *Amanita amerivirosa*. (**B**) *Amanita virosa*. (**C**) *Amanita verna*. (**D**) *Amanita vidua*. (**E**) *Amanita phalloides* var. *alba*. Reproduced with permission of the authors and the editor from Patrick Reumaux and Xavier Carteret, Les Tueurs, published by Klincksieck editions, Paris, 2013 (except D, original drawing by X. Carteret). Bar = 1 cm.

**Figure 4 biology-11-00770-f004:**
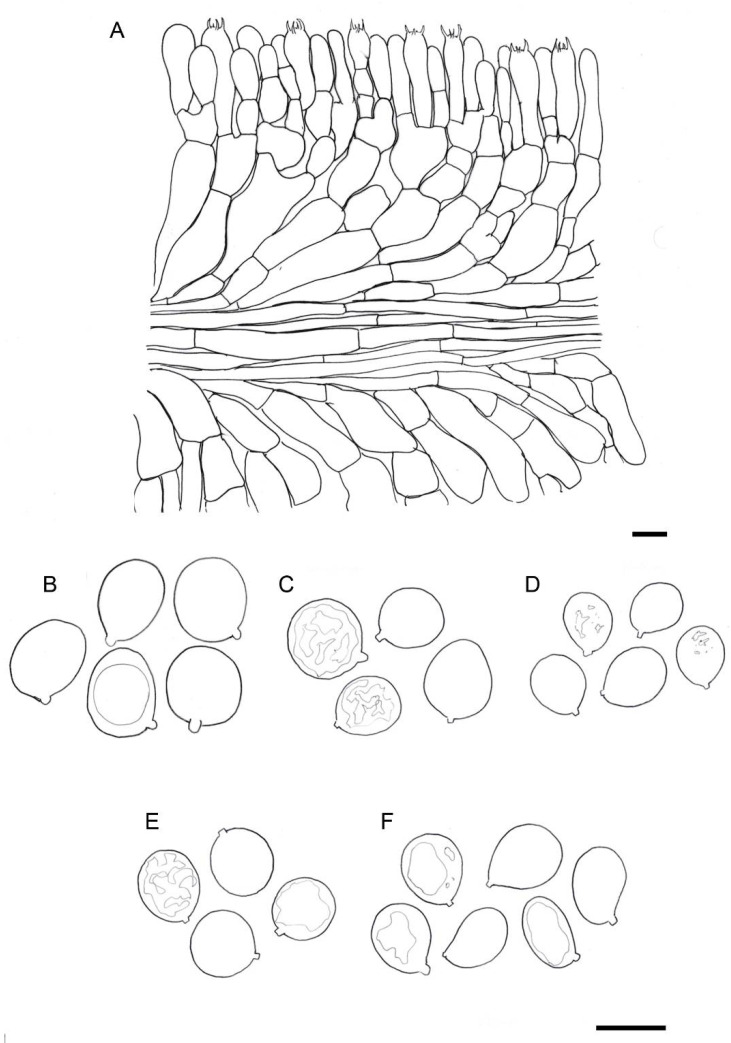
Drawings of microscopical features of destroying angels. (**A**) hymenophoral trama of *Amanita vidua* (LIP:0104591), longitudinal section. (**B**) Basidiospores of *Amanita vidua* (LIP:0401591). (**C**) Basidiospores of *A. amerivirosa* (LIP:0401671). (**D**) Basidiospores of *A. virosa* (LIP:0402243). (**E**) Basidiospores of *A. phalloides* (LIP:0401670). (**F**) Basidiospores of *A. verna* (LIP:0402249). Bar = 10 µm.

**Figure 5 biology-11-00770-f005:**
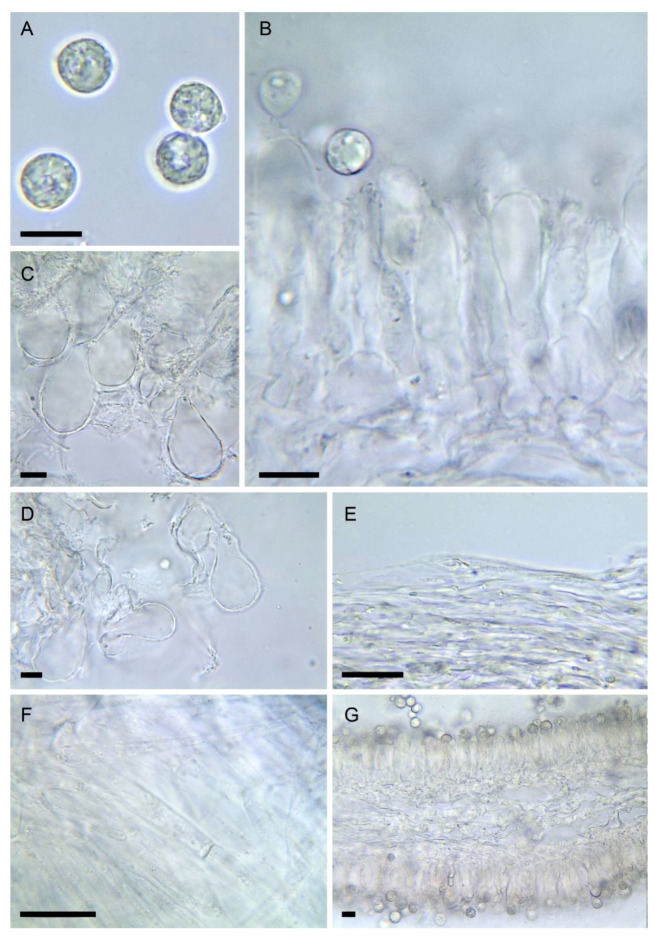
*Amanita amerivirosa* (LIP:0401671). (**A**) Basidiospores. (**B**) Hymenium and subhymenium. (**C**) Lamella edge with terminal cells. (**D**) Elements of partial veil (upper surface). (**E**) Elements of suprapellis. (**F**) Pileipellis, radial section. (**G**) Hymenophore, longitudinal section. All observations in 5% KOH solution. Bar = 10 µm.

**Figure 6 biology-11-00770-f006:**
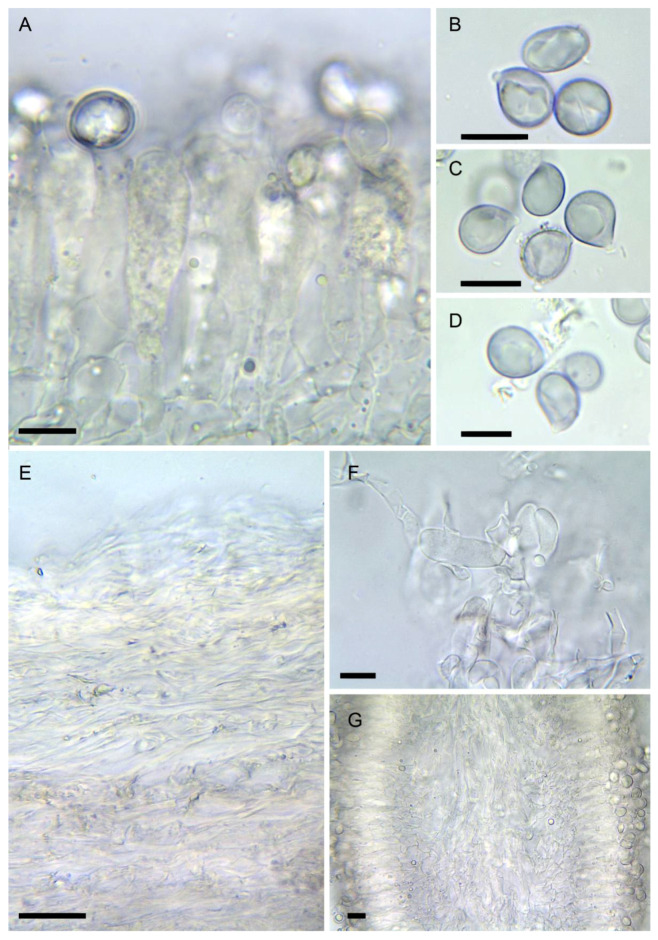
*Amanita verna* (LIP:0402244, except **B**: LIP:0402249. (**A**) Hymenium and subhymenium. (**B**–**D**) Basidiospores. (**E**) Pileipellis, radial section. (**F**) Elements of partial veil (upper surface). (**G**) Hymenophore, longitudinal section. Observations in 5% KOH solution except A, B: Melzer. Bar = 10 µm.

**Figure 7 biology-11-00770-f007:**
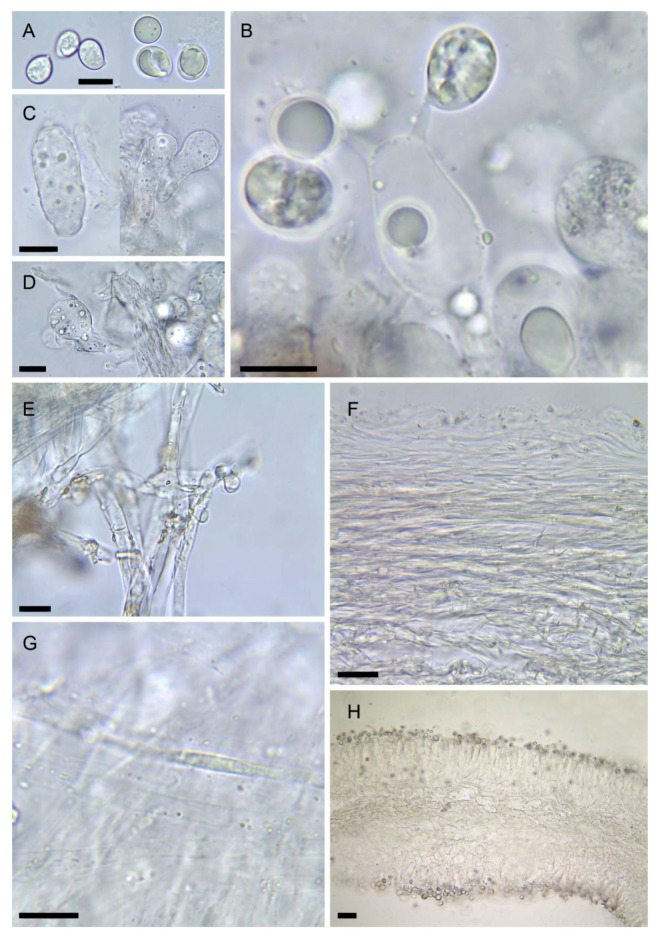
*Amanita vidua* (LIP:0401591). (**A**) Basidiospores. (**B**) Hymenium. (**C**) Terminal cells on lamella edge. (**D**) Elements of partial veil (upper surface). (**E**) Universal veil, outer surface. (**F**) Elements of suprapellis. (**G**) Pileipellis, radial section. (**H**) Hymenophore, longitudinal section. All observations in 5% KOH solution. Bar = 10 µm.

**Figure 8 biology-11-00770-f008:**
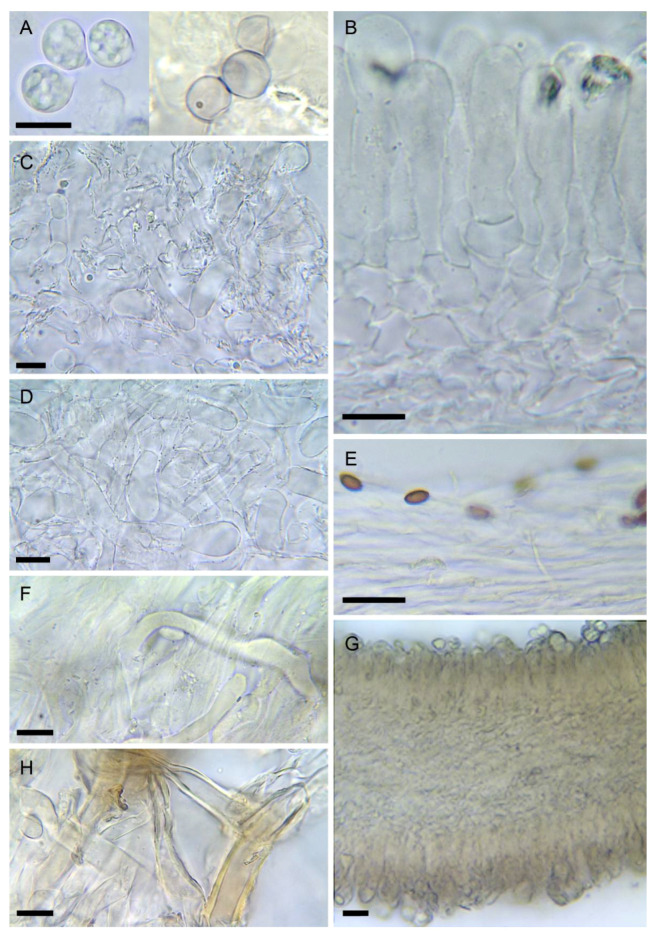
*Amanita virosa* (LIP:0402243). (**A**) Basidiospores. (**B**) Hymenium and subhymenium. (**C**) Elements of lamella edge. (**D**) Elements of partial veil (upper surface). (**E**) Pileipellis, radial section. (**F**) Hyphae in subpellis showing two thromboplerous hyphae. (**G**) Hymenophore, longitudinal section. (**H**) Universal veil, outer surface. All observations in 5% KOH solution except A (left): Melzer. Bar = 10 µm.

**Figure 9 biology-11-00770-f009:**
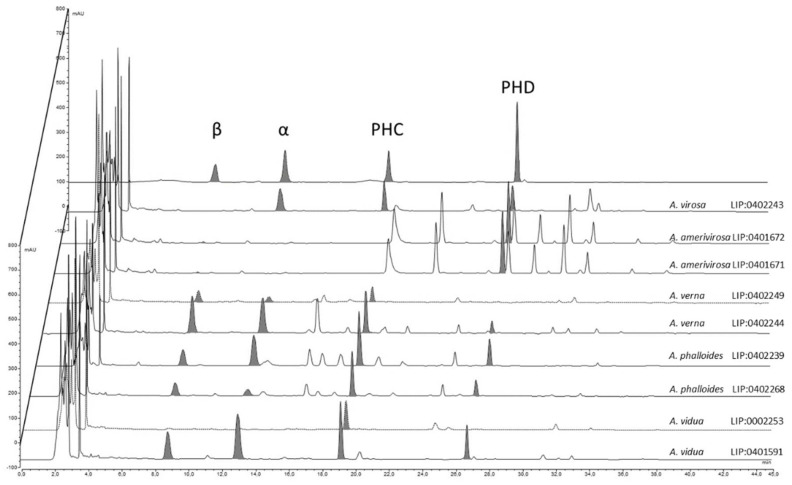
Contents of amatoxins and phallotoxins in the different species of Mediterranean ‘destroying angels’ inferred from phylogenetic analysis: α = α-amanitin, β = β-amanitin, PHC = phallacidin, PHD = phalloidin.

**Figure 10 biology-11-00770-f010:**
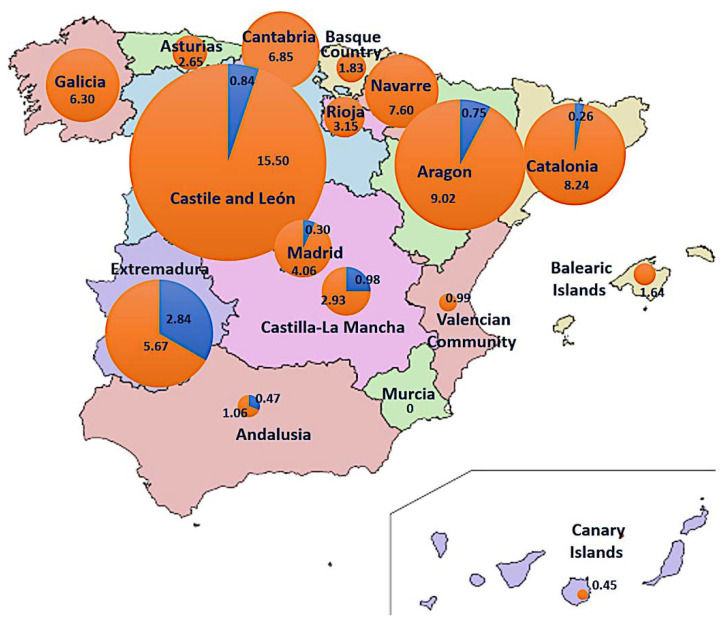
Cumulative incidence of phalloidian poisonings in Spain per region and season (blue: spring, orange: autumn). Values indicate Incidence (per million of habitants) in 1997–2020 period (circumference radius are proportional to the incidence value).

**Table 1 biology-11-00770-t001:** Main interpretations of *Amanita verna* in literature.

Authors	*A. verna* KOH+	*A. verna* KOH-
Bulliard (1783)	Undefined	
Konrad & Maublanc (1934)	*A. phalloides* f. *alba*	
Bataille (1945)	Not cited	*A. verna*
Bertault (1965)	Not cited	*A. verna*
Trimbach (1970)	*A. verna* var. *decipiens*	*A. verna*/*A. verna* var. *tarda*
Romagnesi (1984)	*A. verna*	*A. verna* f. *ellipsospora*
Contu (2000)	*A. decipiens*	*A. verna*/*A. tarda*
Neville & Poumarat (2005)	*A. verna* var. *decipiens*	*A. verna*
**Present study**	** *A. verna* **	** *A. vidua* **

## Data Availability

Not applicable.

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
