# Peer review of "Amanita Section Phalloideae Species in the Mediterranean Basin: Destroying Angels Reviewed"

_biology, 2022, doi:10.3390/biology11050770_

Round 1

Reviewer 1 Report

This manuscript deals with the lethal amanitas from the Mediterranean region based on both phylogenetic analysis of ITS and 28S rDNA sequences and morphological data. It is well-known during the long taxonomic history of the genus Amanita in Europe, some taxonomical controversial issues appeared in the literature. In this work some of the confusions in the taxonomy of lethal amanitas have been clarified, and, thus, it merits publication. The reviewer hopes it will become an important and reliable reference for the future study. However, to reach such a goal, a major revision should be made. The followings should be considered during the improvement.

Line 71: The citation of the occurrence of Amanita phalloides in China [lit. 7] is not encouraged, because this species doesn’t occur in China, based on the last nearly ca. 30 years of observations in the field and literature [lit. 31-33; Yang ZL. 1997, Bibl. Mycol. 170; Chen et al. 2014, Fung. Divers…]. The so called “A. phalloides” in China may well be A. subjunquillea.

The shape and size of basidiospores are usually very important for the recognition of Amanita species. This is certainly the case for the lethal amanitas in the Mediterranean region. Therefore, it is strongly suggested that the authors follow the definitions of Bas (1969, p. 321) strictly and measure the spores carefully in side-view, although DNA sequences can be used for identification but this is mostly only for professional researchers, not for the ordinary readers. The use of the terms by the authors in the present paper is loose. For example, the spores of A. vidua are ”broadly ellipsoid to subglobose, rarely ovoid, … Q = 1.15–1.30 (–1.60)”. If the Q is right, the spores should be mostly broadly ellipsoid, rarely ellipsoid, as that indicated in the brackets.

For A. amerivirosa, spores are “(7.1–) 8.1–10.2 (–11.8) × (6.7–) 7.3–9.3 (–11.1) µm, Q = 1.00–1.20, globose to subglobose”. If so, the spores should be globose to subglobose to broadly ellipsoid. However, this may not be true. The spores of this species are globose to subglobose, infrequently broadly ellipsoid (Tullos et al. 2021).

Although the authors have cited many collections under several of the species dealt in the paper, very limited numbers of collections and spores from each collections have been measured (e.g. lines 324-325, 425-426, 496-498, 632-633, 716-717), and the statistical analyses are consequently often not informative enough. This is not beneficial for the understanding the concept of the lethal amanita there. For example, for the new species of A. vidua, only 28 basidiospores were measured from the holotype, although over 10 collections have been cited. Such treatment might lead additional confusions as those occurred in the European mycological history. For the epitype description of A. virosa, the information provided in lines 716-717 is not accurate enough for the reader. It is stated that “spores subglobose to broadly ellipsoid, (8.1–) 9.0–10.2 × (6.7–) 6.9–9.4 (–9.6) µm, Q = 1.1–1.3 (–1.4)”. And in line 925, “A. virosa (Q=1.00-1.30), lacrymoid to ellipsoid”. This is not the case! Tulloss et al (2021) stated that spores of A. virosa are “[60/3/3] (6.6-) 8.2 - 10.5 (-13.0) × (6.1-) 6.9 - 9.5 (-12.6) µm, (L = 8.9 - 9.5 µm; L’ = 9.3 µm; W = 8.4- 8.6 µm; W’ = 8.6 µm; Q = (1.02-) 1.04 - 1.15 (-1.20); Q = 1.06 - 1.10; Q’ = 1.09), and Cui et al. (2018) stated that basidiospores of A. virosa “[100/9/9] (7.5–) 8.0–11.0 (–12.0) 9 (7.0–)8.0–10.0 (–11.0) lm, Q = 1.0–1.14 (–1.21), Qm = 1.07 ± 0.05, globose to subglobose.”, indicating the spores of A. virosa are in fact globose to subglobose, rarely broadly ellipsoid. The reviewer believes that if the number of spores measured is large enough and statistically analyzed, the shape and the size of the basidiospores of lethal Amanita can be well circumscribed by the data itself and the species concept can be then recognized relatively easily.

Therefore, it is strongly suggested to measure more spores and make accurate and statistical analyses in order to give the reader a reliable and useful information about the shape and size of basidiospores for the species dealt in the paper. Please indicate in the description how many spores from how many collections and how many fruitbodies have been measured. Please provide Q value for each of the species. The authors might think such data can be found in many other works. However, serious scientific accurate data about the size and spores of some common amanitas, like A. phalloides, are largely wanted.

Lines 739-742: This, especially “a strictly Euro-Siberian species”, may provide misleading information to the reader. To the knowledge of the reviewer, the occurrence of A. virosa in northeastern and central parts of China, and Japan has been repeatedly confirmed by DNA sequences and morphological data [lit. 31-33; molecular tree of Tulloss et al. 2021]. This is also indicated in Fig. 1: HMJAU 23304 is A. virosa and was collected from China.

Lines 245-247: It would be helpful to the reader if literature on the sect. Amanita sect. Arenaria can be indicated.

Fig.1: It would be useful if the “epitye” of A. virosa and A. verna is indicated in the phylogram.

Fig. 4: Something must be wrong, e.g. legend of G is absent… Please double-check the figures and their corresponding collections! 

Lines 275-276: “figures 2-4”, and citations in other places like this, are not accurate citation at all.

The collection with cylindrical spores of A. verna is LIP0402242 in line 261, PAM14112205 in lines 497 & 573. Which is right? Both?

line 479: the legend for Fig.6 a-d is hard to understand. Please cite the collections directly and make a double-check.

Lines 762 &771: There must be some mistakes in the key, two “4” are present!

Line 930-935: the thickness of subhymenium for the characterization of the taxa may not be reliable because the thickness can be strongly influenced by the dry or moist niches, and many dried collections can’t be rehydrated well and accurate measurement is often not possible. It is suggested to delete the sentences, because such information may mislead fresh mycologists.

Generally speaking, there are usually no true cheilocystidia in Amanita.

Reviewer 2 Report

In Title, I suggest to replace the "destroying angels reviewed" as "a review of destroying angels in Spain".

In Simple Summary, line 43, please change the "revisited, and a new" to "rivised, with a new"; line 45, change the "discussed" to "was discussed".

In Abstract, line 54, please revised the "such as A. virosa" to "namely A. verna and A. virosa".

In Keywords, pleae delete "Amanita amerivirosa, Amanita decipiens, Amanita phalloides, Amanita porrinensis, Amanita verna, Amanita vidua, Amanita virosa"; please add another keyword "Agaricales".

In Introduction, please provide the taxonomic position for the genus Amanita, that is Amanitaceae, Agaricales, Agaricomycetes, Basidiomycota, Fungi.

In the Supplementary File 1, please give an explanation for the items in bold.

One more question, why do too many samples lack the sequences of tef1 and rpb2?

Reviewer 3 Report

The MS of Alvarado et al. entitled “Amanita section Phalloideae species in the Mediterranean basin: destroying angels reviewed” represent a significant contribution to the proper identification and characterization of these highly poisonous group of mushrooms.  The most notable contribution is the description of a new taxon under the name of A. vidua. The MS is very well written, morphological and biochemical data is well presented. Molecular data, based on the ITS and LSU, is however, only partially convincing. A few protein-coding sequences are still needed to sort better the relationships among A. verna, A. phalloides, the new- described A. vidua, and A subjunquilea. Some other branching in the phylogram, containing other species, were not fully resolved (will be mentioned below). Likely further studies will clarify better the relationship. The authors added some very general information regarding their epidemiology using data restricted to Spain.

We agree that most of the taxa in the Amanita section Phalloideae are described, using the common name, as destroying angels. However, the common name for Amanita phalloides is death cap, or more general, Euro-Asian death cap. The other members described as destroying angels are the European Destroying Angel (Amanita virosa), North American Destroying Angel  (Amanita amerivirosa), European Springtime Destroying Angel (Amanita verna) and the East Asian Death Cap (Amanita subjunquillea). Maybe, for the larger readership, the common names can be mentioned somewhere in the MS as well as the fact that in this MS, when describing the species in the section Phalloideae, the general name of destroying angels is used, in spite of the fact that the common name of some taxa in this group is death cap.

Minor comments:

Line 167.

“ultrasonic waves” could be sonication or ultrasound assisted extraction.

Lines 189 and 190

Λmax. Max should be in subscript.

Lines 233-235.

Amanita virosa and A. amerivirosa/A. virosa var. levipes are significantly related to A. subpallidorosea Hai J. Li, but their phylogenetic relation with A. ocreata Peck, found in the multigenic analysis performed by Codjia et al. [34] could not be recovered with the present analysis.”

Actually, the cluster containing A. virosa, A. amerivirosa, A. subpalidorosea, A. fuliginea, A. rimosa and A. oreata is unrelsolved. What is the support value for this node?

Generally, in figure 1, the support values are extremely hard to read.

Lines 237-238.

“ITS rDNA sequences of A. strobiliformis are 92% similar to those of A. sabulicola S. Morini et al., which is by now the closest relative.“

Was A. sabulicola analyzed in the current study? Could not find it in the phylogram!

Figure 4. Line 268

The bars are missing the required information, that is, the µm!

Lines 329-332 and 340-344.

The same information about habitat and distribution is provided twice. I suggest removing the general info from the note.

Line 432

0,5 should be 0.5

Line 632

It is not clear why the following info was added “[28 spores from the 632 holotype]” after measuring. Were these measurements done using 28 spores from the holotype? If yes, this info could be added after the range of size.

Lines 680-685

Can some molecular info be added about identity/similarity at ITS and LSU between A. vidua and A. verna and A. vidua and A. phalloides be added at this point? Maybe a matrix could be added as supplementary material.

Line 771

There is a mistake in the key. Likely that in line 771, 4 should be 5. 4 was listed already in line 762.

Line 873

Why Ag. bulbosus and not A. bulbosus

Line 959

Why “reaching recently the Paris area [104]. and Paris [102-104].”? Should be only Paris area with all three citations? Or a comma is needed.

Lines 979

“The average incidence of phalloidian poisonings (mostly due to lethal Amanita species) in Spain during the period analyzed in the present work was 0.25 / 1 000 000 inhab per year.”

This is new data, not presented in section 3.4. From where is this data coming, that is, from which of the databases mentioned in the MS? Or is it coming from a published report or paper? 

Inhabit should be inhabitants or inhabit.

Round 2

Reviewer 1 Report

It can be accepted for publication now.